# Empirical prediction for travel distance of channelized rock avalanches in the Wenchuan earthquake area

Weiwei Zhan, Xuanmei Fan, Runqiu Huang, Xiangjun Pei, Qiang Xu, Weile Li

State Key Laboratory of Geohazard Prevention and Geoenvironment Protection, Chengdu University of

- Technology, Chengdu, 610059, China
- *Correspondence to*: Xuanmei Fan (<u>fxm\_cdut@qq.com</u>)

Abstract. Rock avalanches are extremely rapid, massive flow-like movements of fragmented rock. The 10 travel path of the rock avalanches may be confined by channels in some cases, which were named as the 11 channelized rock avalanches. Channelized rock avalanches are potentially dangerous due to their hardly predictable travel distance. In this study, we constructed a dataset with detailed characteristic parameters 12 13 of 38 channelized rock avalanches triggered by the 2008 Wenchuan earthquake using the visual 14 interpretation of remote sensing imagery, field investigation, and literature review. Based on this dataset, 15 we assessed the influence of different factors on the runout distance and developed prediction models of 16 the channelized rock avalanches using the multivariate regression method. The results suggested that the 17 movement of channelized rock avalanche was dominated by the landslide volume, total relief, and 18 channel gradient. The performance of both models was then tested with an independent validation dataset 19 of 8 rock avalanches that induced by the 2008 Wenchuan, the Ms7.0 Lushan earthquake, and heavy 20 rainfall in 2013, showing acceptable good prediction results. Therefore, the travel distance prediction 21 models for channelized rock avalanches constructed in this study is applicable and reliable for predicting 22 the run out of similar rock avalanches in other regions.

23

Keywords: channelized rock avalanches; travel distance; empirical prediction; multivariate regression
 model; Wenchuan earthquake

1 Introduction

Rock avalanches are extremely rapid, massive flow-like movements of fragmented rock from a very
large rock slide or rock fall (Hungr et al. 2014). Hundreds of rapid and long run-out rock avalanches
were triggered by 2008 Wenchuan earthquake in Sichuan Province (Zhang et al. 2013), with catastrophic

consequences for residents in the affected areas. For instance, the  $15 \times 10^6$  m<sup>3</sup> Donghekou rock avalanche 31 in Qingchuan County, near the seismogenic fault, travelled 2.4 km, killing about 780 persons and 32 destroying four villages (Zhang et al. 2013). Rock avalanches can cause incredible damage due to their 33 characteristics of high-speed and unexpectedly long runout, while their transport mechanisms are still 34 considered to be controversial among many researchers (Hungr et al. 2001). Therefore, constructing 35 prediction models for rock avalanche travel distance is meaningful in terms of not only theoretical 36 research on motion mechanisms but also in practical application for risk mitigation of rock avalanches. 37 Methods for determining the travel distance of landslides can be divided into two categories: dynamic 38 modeling (Heim 1932; Sassa 1988; Hungr et al. 2009; Pastor et al. 2009; Lo et al. 2011; ), and empirical 39 modeling (Scheidegger 1973; Lied et al 1980; Corominas, 1996; Finlay et al. 1999; Van Westen et al. 40 2006; Guo et al. 2014). The dynamic models are able to provide information on landslide intensity, such 41 as velocity, affected area and deposition depth, in addition to travel distance. Nonetheless, dynamic 42 models with a variety of physical bases require accurately quantified input parameters that are difficult 43 to obtain before the events, and many simplified assumptions that are not applicable to the actual situation. 44 Recently Mergili et al. (2015) developed a multi-functional open source tool r.randomwalk for 45 conceptual modelling of the propagation of mass movements, which can combine the empirical model 46 with the numerical model. Empirical models considering the correlations between observational data provide an effective technique to aid in understanding mechanisms of rock avalanche motion and to 47 48 develop practical models for predicting rock avalanche travel distance. However, the empirical-statistical 49 models set up from samples with different geomorphological and geological surroundings, trigger 50 conditions, or failure modes are not very sufficient to be applied to the Wenchuan earthquake area.

In this study, we compiled a dataset of 38 rock avalanches with flow paths confined by channels (this 53 kind of landslide is hereinafter termed as channelized rock avalanche) from interpretation of remote 54 sensing, field investigations and literature review (see Section 3.1). Statistical correlations were used to 55 determine the principle factors affecting the mobility of the channelized rock avalanches. Then a stepwise 56 multivariate regression model was developed to build a best-fit empirical model for the travel-distance 57 prediction of this kind of rock avalanches in the Wenchuan earthquake area. A derivative multivariate 58 regression model was also constructed. The performance of both models was then tested with an 59 independent validation dataset of 8 rock avalanches in the same area.

#### 2 Rock avalanches in study area

The study area (see Figure 1) is on the northeast-trending Longmenshan thrust fault zone between the Sichuan basin and the Tibetan plateau. Three major sub-parallel faults are: the Wenchuan-Maowen fault, the Yingxiu-Beichuan fault and the Pengguan fault (Fan et al., 2014). With long-term endogenic and exogenic geological process, this region is characterized by high mountains and deep gorges with extreme rates of erosion (Qi et al 2011).

# [Fig.1 somewhere here]

This study selected 38 channelized rock avalanches induced by the Wenchuan earthquake to study the 70 relations between travel distance and influential factors. These rock avalanches occurred along the 71 seismogenic Yingxiu-Beichuan fault; the distance to the fault ranged from 0 m ~21,300 m with a mean 72 value of 3,895 m. Another distribution characteristic was that these rock avalanches mainly clustered on 73 the step-overs, bends and distal ends of the seismogenic fault. These distribution characteristics of the 74 large rock avalanches suggested that the occurrence of rock avalanches was associated with very strong 75 earthquake ground motion. The Wolong Station recorded the highest seismic acceleration with the peak 76 ground acceleration reaching 0.948g vertically and 0.958g horizontally (Yu et al., 2009). Locally, the 77 ground motion was high enough to throw rocks into the air.

The lithology of outcropping rock in source areas can be divided to four types: carbonate rock, phyllite, igneous rock and sandstone. The deposit of the rock avalanches in the study area was usually debris with mean particle size as tens of centimeters, which suggests that the sliding masses were intensively fragmented during their movement.

The influence of the local geomorphology on the topography of the rock avalanche depositions can be recognized from remote-sensing images after the earthquake. The source area and the transition area of channelized rock avalanches in the study area were somehow easy to be differentiated, as the source area are normally located at the top or upper part of slope, while the flow path (flow or transition area) is partially or fully confined by channels (Figure 2).

# 90 **3 Data and method**

#### 91 **3.1 General consideration**

Various statistical methods have been applied to predict travel distance of landslides, and some popular 93 relationships are summarized in Table 1. The most prevalent one is the equivalent friction coefficient 94 model, which only takes account of landslide volume (Scheidegger, 1973). Another well-known model 95 is the statistical  $\alpha$ - $\beta$  model in which the maximum runout distance is solely a function of topographic 96 conditions (Lied et al., 1980; Gauer et al. 2010). Finlay et al. (1999) developed some multiple regression 97 models containing slope geometric parameters like slope height and slope angle for the travel distance 98 prediction of landslides on the artificial slopes upon the horizontal surface. Based on the data of 54 99 landslides which was relatively open or confined by gentle lateral slope, Guo et al. (2014) established an 100 empirical model for predicting landslide travel distance in Wenchuan earthquake area and suggested that 101 rock type, landslide volume, and slope transition angle (between the failed upper slope and lower slope) 102 play dominant roles on landslide travel distance. And there are increasing sound that the prediction 103 models of travel distance should adapt to different types of landslides (Corominas 1996; Fan et al, 2014).

Table 1 somewhere here

Moreover, the shape and mobility of rock avalanches are controlled by the local topography. Heim (1932) 108 firstly mentioned the influence of local morphology that the debris masses will undergo different effects 109 with the angle of reach changing, and rock avalanches has to conform to the local morphology regardless 110 of their scale. Abele (1974) summarized four different possibilities of adaptation of the rock avalanche 111 to local morphology. Hsu (1975) noted that a sinuous pathway can reduced runout distance of rock 112 avalanches. Nicoletti (1991) inferred that local morphology impacts on landslide motion through 113 changing the rate of total energy dissipation along the travel path. To determine the influence of specific 114 channels on the travel distances of rock avalanches, we respectively consider the impacts of gradients of 115 the upper slopes and lower channels.

Rock avalanches triggered by the Wenchuan earthquake usually initiated from top or the higher part of 118 slopes possibly due to the altitude amplification effect of earthquake acceleration, therefore the toes of 119 the rupture surface were commonly found in the source area at the upstream of the pre-existing channel 120 (See Figure 3). When the slope failed, the failed mass travelled a long distance down the channel. The 121 38 rock avalanches in this study are selected with the criterion that the flow path is partially or fully 122 confined by channels. The volumes of these rock avalanches ranged from  $0.4-50 \times 10^6 \text{ m}^3$ ; with horizontal travel distances between 0.58 and 4.00 km. The volume is prior to the area to be put into the travel 123 124 distance prediction model as it had much more physical meanings. And we introduced total relief as well 125 as the height of source area to probe the influences of the potential energy difference and altitude 126 difference of source mass on the travel distance of the rock avalanches.

# [Fig.3 somewhere here]

3.2 Data

The terms and notations of a typical channelized rock avalanche are shown in Figure 3. The local 130 morphology of a rock avalanche can be divided to three sections: initiated slope (source area), channel 131 (main travel path or flow area) and valley floor (deposition area). When the mass moves over the initiated 132 slope section, it is free from lateral constraints, and the moving mass is able to spread laterally. After 133 entering the channel, the flowing mass is constrained by the two lateral slopes. Finally, the mass may 134 reach to a wide valley floor, where it spreads laterally and deposits. The average inclination of the source 135 area and travel path are obtained respectively, while the gradient of valley floor (deposition area) is 136 neglected as it has very little variation. Slope angle ( $\alpha$ ), denotes the average inclination of the initiated 137 slope section. Channel angle  $(\beta)$ , denotes the average inclination of the sectional channel. Source area 138 height (Hs), denotes the elevation difference between the crest of the sliding source and the toe of the 139 rupture surface. Total relief (H) is the elevation difference between the crest of the sliding source and the 140 distal end of the debris deposit. Travel distance (L) is the horizontal Euclidean distance between the crest 141 of the sliding source and the distal end of the debris deposit. Landslide area (A) is the source area of the 142 rock avalanche obtained from remote sensing image interpretation. An empirical scaling relationship 143 with different empirical coefficients is frequently used to link the volume and the area of landslides in 144 different areas or with different types, and we chose the one developed by Parker et al. (2011) in the 145 same study area. For some rock avalanches with field measured volume available, we use field measurement data rather than the estimated volume by area. The parameters of 38 rock avalanches are
listed in Table 2.

# **Table 2 somewhere here**

#### 150 3.3 Method

Travel distance is the most important prediction parameter in rock avalanche hazard evaluation in 152 mountainous areas. Travel distance prediction of rock avalanche is a complicated issue as it is determined 153 by many different properties of the materials (i.e., grain size distribution and water content), 154 topographical factors, mobility mechanics of failed mass, the confinement attributes of travel path, and 155 so on (Guo et al., 2014). Empirical-statistical methods have long been used as tools to study the mobility 156 of rock avalanche since they are easy to develop and apply, and they are not dependent on knowing the 157 complex physical processes involved in the hypermobility of rock avalanches. Channelized rock 158 avalanches have unique movement paths involving complex, and possibly little-known physical 159 processes such as grain collisions, fragmentation and entrainment of bed material from the channel sides 160 and bottom. Existing empirical models have not produced a favourable prediction. The forecasting index 161 system and the prediction model of channelized rock avalanches should be discussed first.

In this paper, we first selected controlling factors on rock avalanche travel distance through correlation 164 analysis. Then we fitted a stepwise multivariate regression model using all significant correlation 165 variables to obtain a best-fit empirical model for landslide travel distance, and explored which factors 166 were statistically significant at the same time, as expressed in equation (1).

$y = b_0 + b_1 x_1 + b_2 x_2 + b_3 x_3 + \dots + b_n x_n + \varepsilon$ (1)

where *y* is the predictant ('dependent variable'), e.g. travel distance of rock avalanche,  $x_i$  (i = 1, 2, ..., n) are the predictors ('independent variables'),  $b_0$  is the intercept,  $b_i$  (i = 1, 2, ..., n) are the regression coefficients of the corresponding, and  $\varepsilon$  is the residual error, here assumed to be independently and normally distributed. Predictors were added to the regression equation one at a time until there was no significant improvement in parsimonious fit as determined by the adjusted R<sup>2</sup>.

#### 173 **4 Results and validation**

#### 174 **4.1 Reach angle of channelized rock avalanches**

Reach angle, also called the apparent coefficient of friction, is a well-known index to express the landslide mobility. It is the angle of the line connecting the crown of the landslide source area to the toe of the displaced mass. This angle is firstly conducted by Heim (1932) in the famous energy-line model as the average coefficient of friction of a sliding mass from initiation to rest. The reach angle is supposed to possess the ability of landslide mobility prediction because of its tendency to decrease with the increase of landslide volume as illustrated by many researchers (Scheidegger, 1973; Corominas, 1996).

In this study, the influence of landslide volume, drop height, slope of the source area and flow path 183 (channel) on the reach angle of the channelized rock avalanches are examined respectively (Figure 4 and 184 5). Figure 4(a) presents Log(volume) vs. Log(reach angle), showing a weak correlation probably due to 185 the limited volume range in our dataset, constrained movement in channel and local morphology of channels. In order to analyse the effect of potential energy on the reach angle, the effective drop height 186 187 (defined as the total height minus the height of source area) is used instead of the total height to exclude 188 the effect of the superposition of source height and total height. That is especially useful for landslides 189 with large-size initiation but limited travel distance. A significant positive correlation is observed 190 between the reach angle and effective drop height, apart from the four lower scatters in the Figure 4(b). 191 Figure 5(a) and (b) indicate obvious positive correlations between the reach angle with both the slope 192 gradient in source area and channel gradient along the flow path. The large scatter in Figure 4 and 5 193 suggests that the reach angle of channelized rock avalanches might be controlled by some other factors, 194 such as local topography rather than volume, but this needs to be further studied.

- [Fig.4 somewhere here]
- [Fig.5 somewhere here]

# 198 **4.2** Relationships between travel distance and volume, topographic relief of rock avalanche

Correlation coefficients between different variables and travel distance (L) were calculated first,

generating the correlation coefficients matrix shown in Table 3. The significant relevant predictors with

the 95% confidence for travel distance prediction of channelized rock avalanches are landslide area (A),

| 202 | landslide volume (V), total relief (H), source area height (Hs) and channel angle ( $\beta$ ), with correlation |
|-----|-----------------------------------------------------------------------------------------------------------------|
| 203 | coefficient of 0.877, 0.866, 0.857, 0.675, -0.467, respectively.                                                |
| 204 |                                                                                                                 |
| 205 | Table 3 somewhere here                                                                                          |
| 206 | Figure 6 illustrates that the travel distance (L) varies exponentially with the volume (V) of rock avalanche    |
| 207 | with an exponential exponent of 0.377. Compared with a compilation of worldwide rock avalanche data             |
| 208 | (Legros, 2002), the mobility of rock avalanches in our study area is stronger than other non-volcanic           |
| 209 | landslides (power exponent is 0.25), but weaker than volcanic landslides and debris flows (both power           |
| 210 | exponent is 0.39), as shown in Fig.13. The relation between travel distance (L) and total relief (H) is         |
| 211 | shown in Figure 7. The result suggests that the mobility (travel distance) of rock avalanche has relatively     |
| 212 | strong linear relationship with total relief (H). The scale factor is close to 2.4, which means that the        |
| 213 | apparent friction coefficient (H/L) for the rock avalanches is approximately 0.42. This is significantly        |
| 214 | lower than the commonly observed static coefficient of friction of rock material (~0.6).                        |
| 215 | [Fig.6 somewhere here]                                                                                          |
| 216 | [Fig.7 somewhere here]                                                                                          |
| 217 |                                                                                                                 |
|     |                                                                                                                 |

# 218 **4.3 Multivariate regression model of rock avalanche travel distance**

According to the matrix of correlation coefficients (Table 3), the slope angle ( $\alpha$ ) does not have a 220 significant correlation with travel distance (L) at the 95% confidence level. Thus this variable could be 221 excluded first during development of the best-fit regression model for travel distance prediction. Prior to 222 the landslide area (A), the landslide volume (V) has been considered in the models as it has much more 223 physical meaning. In the end, a stepwise linear multivariate regression technique was applied to find the 224 best-fit travel distance regression model using the significant relevant predictors including landslide 225 volume (V), total relief (H), source area height (Hs) and channel angle ( $\beta$ ). The best-fit regression 226 equation for travel distance prediction were derived from the dataset of Table 2 (see equation (2)), and 227 the coefficient of the variables with 95% confidence are shown in Table 4.

$$\log(L) = 0.420 + 0.079 \log(V) + 0.718 \log(H) - 0.365 \log(\tan \beta)$$
(2)

Where log is the logarithm of 10; L is the predicted travel distance (m); V is the landslide volume  $(m^3)$ ;

*H* is the total relief (m);  $\beta$  is the mean gradient of the channel ( ).

Equation (2) can be transformed to equation (3):

$$L = 2.630 V^{0.079} H^{0.718} (\tan \beta)^{-0.365}$$
(3)

The best-fit travel distance regression equation indicates that the travel distance of channelized rock avalanche is positively correlated with landslide scale (landslide volume) and potential energy loss (total relief), and negatively correlated with channel gradient, which is coherent with the results of correlation analysis in Table 3.

While the total relief (*H*) will be unknown prior to landslide occurrence, the elevation difference of source 241 area will be available through specific field investigation on a potential rock avalanche area. Hence, we 242 introduced Hs and  $\alpha$  in replacement of *H* to the regression model as they have relative high correlation 243 with *H* (correlation coefficients are 0.801 and 0.429 respectively). The transformed alternative regression 244 equation is given as equation (4) with the coefficient of the variables with 95% confidence in Table 4.

$$L = 3.6V^{0.303}Hs^{0.244}(\tan\alpha)^{-0.115}(\tan\beta)^{0.072}$$
(4)

Where L is the predicted travel distance (m); V is the landslide volume  $(m^3)$ ; Hs is the height of source 248 area (m);  $\alpha$  is the mean angle of slope segment (?);  $\beta$  is the mean gradient of the channel segment (?). 249 The validity of these two models were evaluated through the significance test leading to the highest  $R^2$ 250 value and the lowest residual standard error. Table 4 shows the significance values for the prediction model equations. Adjusted  $R^2$  means adjusted multiple correlation coefficient, which represents the 251 252 correlation level between the dependent variable and the independent variables. The calculation of adjusted R<sup>2</sup> considers the number of variables and can be used to compare goodness of fit of different 253 254 regression models. Adjusted  $R^2$  of the two regression equations are high, suggesting that the constructed regression models are reliable. The adjusted  $R^2$  of Equation (2) is higher than Equation (4), implying a 255 256 higher precision for the best-fit regression model. The significance test results on the regression equation 257 suggest the significance of multiple regression equations ((F=173.5>  $F_{0.05}(2.883)$ ) for equation (2), and 258  $F=49.5 > F_{0.05}(2.659)$  for equation (4)). Figures 8 (a) and (b) show the distributions of the residuals in

| 259 | relation to the observed travel distance estimated by using equation (2) and (4). Both plots illustrate                                        |
|-----|------------------------------------------------------------------------------------------------------------------------------------------------|
| 260 | normality, constant variance and absence of trends in the residuals.                                                                           |
| 261 |                                                                                                                                                |
| 262 | Table 4 somewhere here                                                                                                                         |
| 263 | Fig.8 somewhere here                                                                                                                           |
| 264 |                                                                                                                                                |
| 265 | Figure 9 compares the predicted travel distances estimated by using equations (2) and (4) with the                                             |
| 266 | observed ones. It suggests that the predicted values of the samples are close to the observed ones. Where                                      |
| 267 | L exceeds 2000 m, the predicted travel distance calculated by using two models are lower than actual                                           |
| 268 | one, with relatively large residual error.                                                                                                     |
| 269 | [Fig.9 somewhere here]                                                                                                                         |
| 270 |                                                                                                                                                |
| 271 | 4.3 Validation                                                                                                                                 |
| 272 | The regression equations were tested using an independent sample validation dataset of 8 rock avalanches                                       |
| 273 | in the same area induced by three different kinds of triggers: 2008 $M_s$ 8.0 Wenchuan earthquake, 2013                                        |
| 274 | $M_s7.0$ Lushan earthquake, and heavy rainfall (Table 5). The volume of these samples ranged from                                              |
| 275 | $88 \times 10^3 - 1.5 \times 10^6 \text{ m}^3$ , and travel distance from $372 - 1372 \text{ m}$ . The background parameters and the predicted |
| 276 | values of each avalanche are listed in Table 5. The relative errors between the predicted values estimated                                     |
| 277 | by using equation (3) and observed values of the travel distance of the rock avalanches,                                                       |
| 278 | $ L_{predicted} - L_{observed} /L_{observed} \times 100\%$ , are between -14.4% and 17.2%, while the relative errors are -44.0%                |
| 279 | and 17.9% for equation (4). On the whole, these two regression models achieved acceptable prediction                                           |
| 280 | accuracy for preliminary forecasting of travel distance of rock avalanches in rugged mountainous areas.                                        |
| 281 | The best-fit regression model appeared to provide greater precision than the alternative model. Regarding                                      |
| 282 | the influence of triggers on the travel distance of the channelized rock avalanches, those triggered by                                        |
| 283 | rainfall and the Lushan earthquake seemed to be more mobile. It is inferred that the former difference is                                      |
| 284 | due to the high water content in failed mass induced by rainfall. A possible reason why two rock                                               |
| 285 | avalanches triggered in the Lushan earthquake travelled farther may be because of structural weakening                                         |
| 286 | of slope rock mass in the 2008 Wenchuan earthquake in the study area.                                                                          |

# [Table 5 somewhere here]

# 288 5 Discussion

#### 289 **5.1 Prediction for travel distance of channelized rock avalanche**

The results of our analysis of the data set, indicates that the mobility (travel distance) of channelized rock 291 avalanche is positively correlated with landslide volume and total relief but negatively correlated with 292 channel gradient. As Figure 6 shows, the travel distance of channelized rock avalanche would rapidly 293 increase with volume of rock avalanche enlarged. Such a high correlation between landslide volume and 294 travel distance implies that the travel distance of channelized rock avalanche is dominated by the 295 spreading of the slide mass (Davies, 1982; Staron, 2009). The high positive correlation between total 296 relief and travel distance is for two reasons: the larger the total relief is, the more kinetic energy the slide 297 mass could obtained and the further distance could it travel (Legros, 2002). The channel gradient is highly 298 correlated with the H/L ratio as shown in Figure 5b, which actually represents the apparent friction 299 coefficient along the flow path similar to the definition of angle of reach by Heim (1932). This is probably 300 the reason of the negative correlation between travel distance and channel gradient, as the decrease of 301 channel gradient means the decrease of static friction coefficient, and the increase of landslide volume 302 and mobility (Figure 4a and Figure 12).

The residual analysis result demonstrates that the projection process in the early motion stage will 305 significantly enlarge the travel distance of rock avalanches. The projection phenomenon was observed 306 in the Wenchuan earthquake region by Huang et al. (2011), defined us the thrown out or projectile motion 307 of slope material due to site amplification effect of seismic wave causing the PGA large than 1 g. The 308 nature of this phenomenon is suggested to be involved with transformation of motion mode from sliding 309 to flowing due to collision and fragmentation effects after the projection (Davies et al, 1999). 310 Furthermore, the degree of fragmentation of failed mass should have remarkable influence on the travel 311 distance of rock avalanche, and other factors changing the fragmentation degree should be further study, 312 such as earthquake effect, geologic structure and rock type.

# 314 **5.2 The mobility of channelized rock avalanches**

The mobility of landslides is influenced by a variety of factors, such as topography, landslide size,

material type, landslide type and water content. The important role of topographical constrains on the

landslide mobility can be referred from the high positive correlation of reach angle with effective drop 318 height, slope gradient and channel gradient (see Figure 4 and 5). Besides, some micro topography like 319 turns (changes of channel flow direction), drop cliff and broad depression along the landslide travel path 320 will influence the motion and deposition of rock avalanches remarkably. The rock avalanches 321 corresponding with the four large bias scatter in Figure 4 (b) are the Wenjia gully, Hongshi Gully, 322 Niumian Gully and Donghekou rock avalanche, whose flow path has cliffs in the upper end of channels 323 with notable drop heights of 260 m, 150 m, 60 m and 160 m respectively according to field investigations. 324 Moreover, fluidization characteristics such as super-elevation near curve transitions can be found in the 325 channel section of these four rock avalanches. This steep micro-topography will enlarge the mobility of 326 rock avalanches because the sliding mass will undergo the drop, collision and fragmentation effects in 327 the early motion stage, which will facilitate motion mode transformation from sliding to flowing. This 328 transformation will enhance the mobility of rock avalanches traveling a much longer distance than 329 predicted. Attention also need to be paid to the broad depression along the channel which is possible to 330 contain a large amount of debris mass and therefore to curb the travel distance of channelized rock 331 avalanches. For example, in the Wenjia Gully almost half of the total volume of the rock avalanche was 332 deposited at the beginning of the channel (see Figure 10(c)), leading to a lower travel distance than 333 expected.

#### [Fig.10 somewhere here]

To investigate the influence of landslide types on the landslide mobility, we compile our dataset with the 338 dataset created by Guo et al. (2014), as it contains the data of 32 landslides with other types (debris 339 avalanches, rock slides, soil slides) triggered by the Wenchuan earthquake. We plot the relationship 340 between L with V and H respectively for different landslide types (see Figure 11 a and b). As shown in 341 Figure 11, rock avalanches have the strongest mobility while soil slides show the weakest one, and the 342 mobility of rock slides is approximate to the mobility of debris avalanches. While compared with the 343 worldwide datasets by using the reach angle as the mobility index (see Figure 12 and 13), our dataset 344 shows a consist tendency with the worldwide datasets presented by Corominas (1996) and Legros (2002). 345 Our dataset could contribute to the worldwide database by filling the gap of rock avalanches.

[Fig.11 somewhere here]
[Fig.12 somewhere here]
[Fig.13 somewhere here]
The common triggers of landsides are earthquakes and rainfall. T

The common triggers of landsides are earthquakes and rainfall. The influence of triggers on landslide 352 distribution has been well studied, but the effect of triggers on the landslide mobility is still a scientific 353 gap. Zhang et al. (2013) indicated that rock avalanches triggered by earthquakes have a slightly lower 354 mobility than ones not triggered by earthquakes, and rock avalanches close to the seismic fault do not 355 always have a higher mobility even when a rock avalanche near the seismic fault is subjected to higher 356 ground accelerations. Guo et al. (2014) also mentioned that the seismic acceleration has less influence 357 than rock type, sliding volume, slope transition angle and slope height on landslide travel distance. 358 According to Table 5, two rainfall-induced rock avalanches show stronger mobility than earthquake-359 induced ones. The rock avalanches induced by rainfall express a stronger mobility than the earthquake-360 induced ones may due to lubrication effect of water However, detailed study on the influence of triggers 361 on the landslide mobility need further dataset.

# 363 6 Conclusion

Channelized rock avalanche refers to a rock avalanche with a flow path confined between valley walls. 365 Relevant detailed data on thirty-eight channelized rock avalanches triggered by Wenchuan earthquake 366 were collected by remote sensing, field investigation and literature review. The results of correlation and 367 regression analysis revealed that the movement of channelized rock avalanches is dominated by 368 spreading of the failed mass. Landslide volume (V), total relief (H) and channel angle ( $\beta$ ) had 369 predominant effects played a dominating role in the on travel distance of channelized rock avalanches. 370 Stepwise multivariate regression was used to develop a nonlinear best-fit travel distance prediction model 371 for the channelized rock avalanches. An alternative multivariate regression model was also built. The 372 reliability of the two models was tested on by an independent validation dataset of 8 rock avalanches in 373 the same area and produced good results, meeting the requirements for preliminary evaluation of travel 374 distance for channelized rock avalanches in the Wenchuan earthquake area.

#### 375 Acknowledgement

- This work was supported by the Fund for International Cooperation (NSFC-RCUK\_NERC), Resilience
- to Earthquake-induced landslide risk in China (Grant No. 41661134010), the Young Scientists Fund of
- the National Natural Science Foundation of China (Grant No. 41302241), the Fund for Creative Research
- Groups of China (Grant No. 41521002), The authors thank Dr. Mauri McSaveney for his constructive
- comments and editing the English the paper.

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

| Approach       | Keywords to                                                                          | Landslide types                                             | Triggers            | Main references                           |
|----------------|--------------------------------------------------------------------------------------|-------------------------------------------------------------|---------------------|-------------------------------------------|
|                | characterize the<br>methods                                                          |                                                             | 88                  |                                           |
| Reach<br>angle | Log H/L=C1Log V+C0                                                                   | Rock<br>fall/slide/avalanche<br>and flow-like<br>landslides | Unkonwn             | Scheidegger, 1973;<br>Corominas,1996      |
|                | $H/L=C_1 \tan S + C_0$                                                               | Soil slides, snow avalanches                                | Non-<br>seismic     | Hunter et al., 2003;<br>Lied et al., 1980 |
| Travel         | Log L=C <sub>1</sub> Rt+C <sub>2</sub> Log V+<br>C <sub>3</sub> sin S+C <sub>0</sub> | Rock/soil slides and<br>rock/debris<br>avalanches,          | Seismic             | Guo et al., 2014                          |
| distance       | $Log L=C_1 Log H+C_2$<br>$Log tanS+C_0$                                              | Soil landslides on artificial slopes                        | Human<br>activities | Finlay et al., 1999                       |
|                | $L=C_1V^{C_2}$                                                                       | Debris slides, debris<br>slides                             | Rainfall            | Jaiswal et al., 2011                      |

**Table 1** Summarization of statistical relationships indicating landslide mobility in the literature

Note: C0, C1, C2, C3 are the constants. L is the travel distance. H is the total height. V is the volume. S
is the average slope angle while St is the slope transition angle. Rt is the rock type.

| Code | Landslide name | Longitude,<br>(E) | Latitude,<br>( N) | Landslide<br>area,<br>A (m <sup>2</sup> ) | Landslide<br>volume,<br>V (m <sup>3</sup> ) | Source<br>area<br>height,<br><i>Hs</i> (m) | Slope angle, $\alpha$ (°) | Channel angle, $\beta$ (°) | Total<br>relief,<br><i>H</i> (m) | Travel<br>distance,<br><i>L</i> (m) | Reference       |
|------|----------------|-------------------|-------------------|-------------------------------------------|---------------------------------------------|--------------------------------------------|---------------------------|----------------------------|----------------------------------|-------------------------------------|-----------------|
| 1    | Wenjia Gully   | 104.140           | 31.552            | 3000566                                   | 50000000                                    | 440                                        | 26                        | 7                          | 1320                             | 4000                                | Xu et al., 2009 |
| 2    | Shuimo Gully   | 103.981           | 31.442            | 915608                                    | 19960000                                    | 490                                        | 35                        | 10                         | 860                              | 2000                                |                 |
| 3    | Dawuji         | 104.196           | 31.702            | 792190                                    | 16330000                                    | 540                                        | 29                        | 13                         | 880                              | 1900                                |                 |
| 4    | Donghekou      | 105.113           | 32.410            | 1283627                                   | 15000000                                    | 240                                        | 25                        | 11                         | 640                              | 2400                                | Xu et al., 2009 |
| 5    | Hongshigou     | 104.130           | 31.624            | 687520                                    | 13410000                                    | 290                                        | 37                        | 17                         | 1040                             | 2700                                |                 |
| 6    | Woqian         | 104.964           | 32.308            | 695672                                    | 12000000                                    | 330                                        | 30                        | 10                         | 560                              | 1600                                | Xu et al., 2009 |
| 7    | Xiaojiashan    | 104.038           | 31.465            | 465899                                    | 7810000                                     | 480                                        | 48                        | 24                         | 930                              | 1350                                |                 |
| 8    | Niumian Gully  | 103.456           | 31.044            | 527700                                    | 7500000                                     | 320                                        | 32                        | 13                         | 800                              | 2640                                | Xu et al., 2009 |
| 9    | Liqi Gully     | 105.207           | 32.169            | 355113                                    | 5360000                                     | 360                                        | 37                        | 12                         | 650                              | 1500                                |                 |
| 10   | Caocaoping     | 104.139           | 31.607            | 354046                                    | 5340000                                     | 345                                        | 31                        | 17                         | 580                              | 1340                                |                 |
| 11   | Huoshi Gully   | 104.134           | 31.616            | 322155                                    | 4680000                                     | 270                                        | 38                        | 17                         | 700                              | 1320                                |                 |
| 12   | Shibangou      | 105.090           | 32.419            | 496983                                    | 4500000                                     | 450                                        | 34                        | 9                          | 650                              | 1800                                | Xu et al., 2009 |
| 13   | Xiejiadianzi   | 103.841           | 31.298            | 294256                                    | 4000000                                     | 400                                        | 34                        | 15                         | 720                              | 1600                                | Xu et al., 2009 |
| 14   | Dashui Gully   | 103.675           | 31.199            | 241874                                    | 3150000                                     | 320                                        | 30                        | 17                         | 560                              | 1400                                |                 |
| 15   | Changping      | 103.754           | 31.259            | 224645                                    | 2840000                                     | 290                                        | 37                        | 16                         | 500                              | 1200                                |                 |
| 16   | Xiaomuling     | 104.102           | 31.613            | 218704                                    | 2740000                                     | 175                                        | 45                        | 26                         | 710                              | 1025                                |                 |
| 17   | Baishuling     | 104.385           | 31.807            | 208968                                    | 2570000                                     | 335                                        | 36                        | 20                         | 620                              | 1200                                |                 |
| 18   | Dawan          | 104.536           | 31.907            | 203959                                    | 2480000                                     | 220                                        | 28                        | 20                         | 480                              | 1000                                |                 |
| 19   | Xiaojiashan    | 104.182           | 31.486            | 198165                                    | 2385499                                     | 340                                        | 44                        | 20                         | 650                              | 1135                                |                 |

Table 2 Data of various factors for establishment of prediction model of rock avalanche travel distance

| 20 | Shicouzi       | 104.918 | 32.243 | 169540 | 1920000 | 260 | 30 | 26 | 640  | 1200 |  |
|----|----------------|---------|--------|--------|---------|-----|----|----|------|------|--|
| 21 | Changtan       | 104.133 | 31.508 | 151094 | 1640000 | 400 | 33 | 25 | 1050 | 1650 |  |
| 22 | Hongmagong     | 104.962 | 32.301 | 144683 | 1540000 | 195 | 30 | 14 | 330  | 800  |  |
| 23 | Baiguocun      | 105.088 | 32.385 | 139800 | 1470000 | 165 | 26 | 12 | 260  | 800  |  |
| 24 | Qinglongcun    | 105.036 | 32.342 | 134079 | 1390000 | 90  | 21 | 11 | 200  | 600  |  |
| 25 | Pengjiashan    | 104.546 | 31.930 | 127156 | 1290000 | 200 | 33 | 28 | 580  | 1000 |  |
| 26 | Longwancun     | 104.571 | 31.922 | 99821  | 920000  | 205 | 31 | 28 | 460  | 860  |  |
| 27 | Zhangzhengbo   | 105.017 | 32.333 | 99726  | 920000  | 125 | 29 | 15 | 320  | 800  |  |
| 28 | Dujiayan       | 105.028 | 32.336 | 94769  | 860000  | 100 | 33 | 17 | 400  | 880  |  |
| 29 | Madiping       | 104.996 | 32.355 | 94632  | 860000  | 140 | 27 | 31 | 395  | 740  |  |
| 30 | Yandiaowo      | 105.099 | 32.391 | 92128  | 820000  | 145 | 30 | 26 | 390  | 800  |  |
| 31 | Chuangzi Gully | 104.085 | 31.518 | 91717  | 820000  | 185 | 35 | 15 | 295  | 670  |  |
| 32 | Zhaojiashan    | 105.041 | 32.342 | 82329  | 700000  | 115 | 22 | 16 | 280  | 700  |  |
| 33 | Weiziping      | 105.083 | 32.387 | 74661  | 620000  | 135 | 22 | 18 | 240  | 600  |  |
| 34 | Maochongshan 2 | 104.908 | 32.243 | 70251  | 570000  | 160 | 38 | 22 | 500  | 740  |  |
| 35 | Waqianshan     | 105.049 | 32.376 | 70007  | 560000  | 135 | 24 | 18 | 250  | 620  |  |
| 36 | Muhongping     | 104.982 | 32.291 | 68288  | 540000  | 175 | 28 | 20 | 420  | 970  |  |
| 37 | Dapingshang    | 104.542 | 31.889 | 65700  | 520000  | 160 | 34 | 29 | 365  | 640  |  |
| 38 | Liushuping 2   | 105.054 | 32.365 | 54810  | 400000  | 150 | 29 | 16 | 240  | 580  |  |

|        | Α     | V     | н     | Hs    | a      | β      | L      |
|--------|-------|-------|-------|-------|--------|--------|--------|
| A      | 1.000 | 0.982 | 0.674 | 0.521 | -0.119 | -0.524 | 0.877  |
| v      |       | 1.000 | 0.713 | 0.560 | -0.055 | -0.492 | 0.866  |
| Н      |       |       | 1.000 | 0.801 | 0.429  | -0.130 | 0.857  |
| Hs     |       |       |       | 1.000 | 0.399  | -0.323 | 0.675  |
| α      |       |       |       |       | 1.000  | 0.264  | 0.082  |
| ß      |       |       |       |       | _      | 1.000  | -0.467 |
| ,<br>L |       |       |       |       | _      | _      | 1.000  |
|        |       |       |       |       |        |        |        |

Table 3 Correlation coefficients of continuous variables listed in Table 2

Note: The number in Italics indicates the two variables are not significantly correlated

|               | •         | 001 1            | 1. 1.      | C ' 'C'           |               | 1              | •               | 1 1  |
|---------------|-----------|------------------|------------|-------------------|---------------|----------------|-----------------|------|
| Toble / The r | arraggion | conttiniante ano | roculte of | t cianiticonco to | to ot two m   | 11 tivorioto 1 | orraggion mag   | 1010 |
|               | CALCSNIOL | COCHICICHINA AIR | г ісзина о | Л МУШНСАНСС ІСА   | MS OF LWO III | טוטעאוואול ו   | CYTESSION IIIOU | 1018 |
|               |           | •••••••••••••••  | 1000100 01 |                   |               |                | •               |      |

| Equations   | Coeffici<br>ents* | Interce<br>pt | Coefficient<br>of log(V) | Coefficient<br>of log(H) | Coefficient of $log(tan\beta)$ | Coefficient<br>of log(Hs) | Coefficient of $log(tan\beta)$ | Adjust<br>ed R <sup>2</sup> | F-stat | F0.05 |
|-------------|-------------------|---------------|--------------------------|--------------------------|--------------------------------|---------------------------|--------------------------------|-----------------------------|--------|-------|
| Best-fit    | LCI               | 0.175         | -0.013                   | 0.521                    | -0.548                         | _                         | _                              |                             |        |       |
| regression  | Mean              | 0.420         | 0.079                    | 0.718                    | -0.365                         | _                         | _                              | 0.933                       | 173.5  | 2.883 |
| equation    | UCI               | 0.665         | 0.171                    | 0.914                    | -0.182                         | _                         | _                              |                             |        |       |
| Alternative | LCI               | 0.110         | 0.199                    | —                        | -0.165                         | -0.002                    | -0.464                         |                             |        |       |
| regression  | Mean              | 0.561         | 0.303                    | _                        | 0.072                          | 0.244                     | -0.115                         | 0.840                       | 49.5   | 2.659 |
| equation    | UCI               | 1.012         | 0.407                    | _                        | 0.308                          | 0.489                     | 0.233                          |                             |        |       |

Note: "Coefficients" of each variable has three kinds: LCI is lower bound of the coefficients with 95% confidence; Mean is the mean value of the coefficients; UCI is upper bound of the coefficients with 95% confidence;

| Landslide<br>name | Longitude | Latitude | Triggers* | V<br>/10 <sup>4</sup> m <sup>3</sup> | α<br>/° | B<br>/ ° | Hs<br>/m | H<br>/m | L<br>/m | L'(3)**<br>/m | Error<br>/ % | L '(4)***<br>/m | Error<br>/ % |
|-------------------|-----------|----------|-----------|--------------------------------------|---------|----------|----------|---------|---------|---------------|--------------|-----------------|--------------|
| Pianqiaozi        | 104.370   | 31.822   | WCEQ      | 8.8                                  | 35      | 19       | 153      | 205     | 372     | 436           | 17.2         | 373             | 0.3          |
| Yangjiayan        | 104.328   | 31.755   | WCEQ      | 25.4                                 | 41      | 23       | 164      | 304     | 518     | 583           | 12.5         | 518             | 0.1          |
| Shanshulin        | 103.508   | 31.181   | WCEQ      | 27.9                                 | 34      | 25       | 340      | 433     | 715     | 731           | 2.3          | 660             | -7.6         |
| Fuyangou          | 103.501   | 31.422   | WCEQ      | 71.9                                 | 38      | 28       | 385      | 530     | 763     | 869           | 13.8         | 900             | 17.9         |

Table 5 Background parameters and predicted values of 8 rock avalanches in the same area used for validation

| Dayanbeng1 | 102.762 | 30.179 | LSEQ | 100 | 53 | 10 | 254 | 424 | 1267 | 1136 | -10.3 | 781 | -38.4 |
|------------|---------|--------|------|-----|----|----|-----|-----|------|------|-------|-----|-------|
| Dayanbeng2 | 102.761 | 30.178 | LSEQ | 110 | 50 | 8  | 237 | 407 | 1372 | 1208 | -12.0 | 787 | -42.6 |
| Ermanshan  | 102.739 | 29.322 | RF   | 100 | 33 | 15 | 148 | 635 | 1370 | 1303 | -4.9  | 767 | -44.0 |
| Wulipo     | 103.567 | 30.919 | RF   | 150 | 30 | 10 | 135 | 377 | 1260 | 1078 | -14.4 | 833 | -33.9 |

Note: "Triggers" is the triggering condition of rock avalanches: "WCEQ" represents the 2008 Wenchuan  $M_s$  8.0 earthquake; "LSEQ" represents the 2013 Lushan  $M_s$  7.0 earthquake; "RF" represents the rock avalanche was induced by heavy rainfall.  $L'_{(3)}$ ,  $L'_{(4)}$  indicates the predicted travel distance estimated by using equation (3) and (4) respectively.