# Peer review of "Empirical prediction for travel distance of channelized rock avalanches in the Wenchuan earthquake area"

_Natural Hazards and Earth System Sciences, 2016_

## Referee Comment (RC3)

**Review of the manuscript "Empirical prediction for travel distance of channelized rock avalanches in the Wenchuan earthquake area" submitted by Weiwei Zhan et al.**

The paper demonstrates an interesting approach to relate the travel distance of channelized rock avalanches to the release volume and to a set of topographic parameters. The purpose of this exercise is to facilitate forward calculation of possible future processes where the release area and volume are known. The research is generally well described, and the manuscript is well structured and illustrated. As it is usual for discussion papers, there is some potential for improvement. I have identified a number of minor to moderate issues that have to be addressed before I can finally recommend the manuscript for publication in NHESS. **All in all, I suggest minor revisions**.

**General comments:**

Pages 3 and 4 are almost identical – I think that page 4 can just be deleted.

A reference that could be interesting:

Mergili, M., Krenn, J., Chu, H.-J. (2015): r.randomwalk v1, a multi-functional conceptual tool for mass movement routing. Geoscientific Model Development 8: 4027-4043. doi:10.5194/gmd-8-4027-2015

**Specific comments:**

Even though the paper is well written in general, there are several minor errors of grammar and style. It would be out of scope to address these shortcomings in detail, therefore I recommend careful copy editing. In the following, I focus on issues concerning the scientific content of the manuscript. The numbers refer to the manuscript lines:

119: "topography" would be suitable rather than "geography"

124: please explain what you mean with "slope transition angle"

130: what is the "angle of impact"?

145: In many cases it is probably hard to clearly delineate the source area from the transition area – maybe you could shortly explain which strategy you applied to do so?

148–165: This part does NOT describe the data you use, but defines some terms. It should be moved to the introduction.

159–160: Is L the Euclidean distance, or the distance along the flow path?

176: You should give some examples or references demonstrating that the existing models did not produce a favourable prediction.

182: You have to explain wat "x" is in Eq. 1.

238: Eq. 5 does not exist.

261: better use $10^3$ or $10^6$ instead of $10^4$.

296: What do you mean here with "projection"?

I hope that my comments will help to further improve the quality of the manuscript. The authors should feel free to contact me in case they disagree with my comments or seek discussion: martin.mergili@univie.ac.at.

With best regards, Martin Mergili

---

## Referee Comment (RC1) · H.-B. Havenith (Referee) · 5 Dec 2016

Dear editor, dear authors,

I read this paper with great interest. However, I think some essential aspects about the ratio of volume versus sliding surface are missing in the discussion and conclusions. You mainly considered the relatively classical parameters of volume (alone), slope angle, and total relief. The problem is treated as being almost 1D (linear along the slope) while channeling of rock avalanches is certainly also depending on the channel cross-section and the presence of 'turns' along the channel. Those two aspects should be analysed as well.

[Figure]

See more detailed comments and minor corrections in the annotated reviewed manuscript.

sincerely yours. Hans-Balder Havenith

Please also note the supplement to this comment:
http://www.nat-hazards-earth-syst-sci-discuss.net/nhess-2016-372/nhess-2016-372-RC1-supplement.pdf

---

## Referee Comment (RC2) · T.ÂăW.ÂăJ. van Asch (Referee) · 11 Jan 2017

T.ÂăW.ÂăJ. van Asch (Referee)

t.w.j.vanasch@uu.nl

This is an interesting paper showing that with a limited amount of factors one is able to predict the travel distance of rock avalanches provided that they occur in the same area, are of the same type and have the same triggering conditions. This was already shown in this paper where the validation with landslides with other triggering conditions and lying in another area gave sometimes poor results I am wondering why the authors did not mention in the introduction explicitly the use of the energy concept for runout modelling, which gives a simple transparent insight in the most important factors ( relief and friction) influencing run-out distance Interesting question arises also from the introduction about advantages and disadvantages of the use of deterministic physical models and statistical models. In the introduction the authors mention examples of important fast landslides but they must more precisely describe triggering condition and type I have great difficulty in presenting the total height (H) as an important factor for the run out distance since it is highly correlated with run-out distance (L) Therefore Equation 2 and 3 are really not useful predictive equations because you need the travel distance L which you have to predict? May be a trial an error procedure for L is a solution when using this equation? It would be nice to test this. The authors solved the problem by making a correlation of Hs with H (Eq 4) which is a practical solution but has of course no physical meaning and it has to be questioned whether it works in other areas. I want to see comments on this in the discussion paragraph. The energy approach to model run-out (not used in this paper) shows that volume does not play a role. But in that case it is assumed that friction is not influenced by volume, which in practice seems to be the case due to all kinds of physical processes in the mass. Therefore in order to show this, I asked the authors to make also a correlation between H/L (mean friction during run-out) and volume. The effect of slope angle beta is a bit strange In Eq, 2 and 3 it is negative while in Eq 4 it is a positive factor. The authors should comment on this. The authors give sometimes unclear and peculiar explanations of their findings regarding the effect of volume on travel distance and the effect of total height and channel angle on run-out distance. A lack of clarity for me sometimes occurred in the text where the authors give no definitions of some terms like flow capacity, projectile motion etc., ( see my annotations and comments) The English is fine but a final check is necessary to make some small corrections (see some of my annotations) I think the authors are able without much difficulty to correct some typos, rephrase some sentences and give comments on my general remarks and annotations. Therefore I recommend minor revision

Please also note the supplement to this comment:
http://www.nat-hazards-earth-syst-sci-discuss.net/nhess-2016-372/nhess-2016-372-RC2-supplement.pdf

[Figure]

**Supplement:**

# Empirical prediction for travel distance of channelized

# rock avalanches in the Wenchuan earthquake area

Weiwei Zhan, Xuanmei Fan, Runqiu Huang, Xiangjun Pei, Qiang Xu, Weile Li

[revised manuscript text omitted]

Sichuan basin and the Tibetan plateau. Three major sub-parallel faults are: the Wenchuan-Maowen fault, the Yingxiu-Beichuan fault and the Pengguan fault (Fan et al., 2014). With highly developed stream systems, this region is characterized by high mountains and deep valleys and extreme rates of erosion (Fu et al 2009; Qi et al 2011).

This study selected 38 channelized rock avalanches induced by the Wenchuan earthquake to study the relations between travel distance and influential factors. These rock avalanches occurred along the seismogenic Yingxiu-Beichuan fault; the distance to the fault ranged from 0 m ~21,300 m with a mean value of 3,895 m. Another distribution characteristic was is that these rock avalanches mainly clustered on the step-overs, bends and distal ends of the seismogenic fault. These distribution characteristics of the large rock avalanches suggested that the occurrence of rock avalanches was associated with very strong earthquake ground motion. The Wolong Station recorded the highest seismic acceleration with the peak ground acceleration reaching 0.948g vertically and 0.958g horizontally (Yu et al., 2009).

Locally, the ground motion was high enough to throw rocks into the air.

The lithology of outcropping rock in source areas can be divided to four types: carbonaterock, phyllite, igneous rock and sandstone. The landslide deposit of the rock avalanches in the study area structure was is usually debris, which suggests that the sliding masses were intensively fragmented during their movement.

The influence of the local geomorphology on the paths of the rock avalanches was obtained from remote-sensing images after the events. Although the rock avalanches we chose all had flow paths confined by channels, some topographic differences were found to be significant in affecting present that had affected the shape morphology of the rock avalanche deposits. The source areas had well- defined boundaries. When the source mass was detached from the slide bedbedrock, it may directly move into the channel down slope (see Figure 2b), or access the channel with enter it at some impact transition angle of movement direction (see Figure 2a).  The channel itself may have changes in direction and inclination. The distal end of the landslide may lie stop in the channel (see Figure 2a) or may reach the wide valley or plain (see Figure 2b).

[Figure]

[Figure]

The study area (see Figure 1) is on the northeast-trending Longmenshan thrust fault zone between the

Sichuan basin and the Tibetan plateau. Three major sub-parallel faults are: the Wenchuan-Maowen fault, the Yingxiu-Beichuan fault and the Pengguan fault (Pan et al., 2014). With highly developed stream systems, this region is characterized by high mountains and deep valleys and extreme rates of erosion (Fu et al 2009; Qi et al 2011).

This study selected 38 channelized rock avalanches induced by the Wenchuan earthquake to study the relations between travel distance and influential factors. These rock avalanches occurred along the seismogenic Yingxiu-Beichuan fault; the distance to the fault ranged from 0 m ~21,300 m with a mean value of 3,895 m. Another distribution characteristic is that these rock avalanches mainly clustered on the step-overs, bends and distal ends of the seismogenic fault. These distribution characteristics of the large rock avalanches suggested that the occurrence of rock avalanches was associated with very strong earthquake ground motion. The Wolong Station recorded the highest seismic acceleration with the peak ground acceleration reaching 0.948g vertically and 0.958g horizontally (Yu et al., 2009). Locally, the ground motion was high enough to throw rocks into the air.

The lithology of outcropping rock in source areas can be divided to four types: carbonaterock, phyllite, igneous rock and sandstone. The deposit of the rock avalanches in the study area is usually debris, which suggests that the sliding masses were intensively fragmented during their movement.

The influence of the local geomorphology on the paths of the rock avalanches was obtained from remote-sensing images after the events. Although the rock avalanches we chose all had flow paths confined by channels, some topographic differences were found to be significant in affecting the morphology of the rock avalanche deposits. When the source mass was detached from the bedrock, it may directly move into the channel down slope (see Figure 2b), or enter the channel with some transition angle of movement direction (see Figure 2a). The channel itself may have changes in direction and inclination. The distal end of the landslide may stop in the channel (see Figure 2a) or may reach to wide valley or plain (see Figure 2b).

[Figure]

**3 Data and method**

**3.1 General consideration**

[revised manuscript text omitted]

Moreover, fluidization characteristics such as super-elevation near curve transitions can be found in the channel section of these four rock avalanches. These findings manifest the steep micro-geotopography will enlarge the mobility of rock avalanches as this kind of topography will lead the slide mass to undergo the projection, collision, fragmentation effects in the early motion stage which will facilitate

motion mode transformation from sliding to flowing. This transformation will enhance the motion mobility of rock avalanche to travel a much longer distance than predicted one.

**4.3 Validation**

The regression equations were tested using an independent sample validation data set (Table 4)of 8

rock avalanches in the same area induced by three different kinds of triggers: 2008 $M_s$7.8 Wenchuan earthquake, 2013 $M_s$7.0 Lushan earthquake, and heavy rainfall. The volume of these samples ranged from $8.8\times10^4$–$150\times10^4$m$^3$, and travel distance from 372–1372 m. The background parameters and the predicted values of each avalanche are listed in Table 4. The relative errors between the predicted values estimated by using equation (3) and observed values of the travel distance of the rock avalanches, |Lpredicted−Lobserved|/Lobserved×100%, are between -14.4% and 17.2%, while the relative errors are -44.0% and 17.9% for equation (4). On the whole, these two regression models achieved acceptable prediction accuracy for preliminary forecasting of travel distance of rock avalanches in rugged mountainous areas. The best-fit regression model appeared to provide greater precision than the alternative model. Regarding the influence of triggers on the travel distance of the channelized rock avalanches, those triggered by rainfall and the Lushan earthquake seemed to be more mobile. It is inferred that the former difference is due to the high water content in failed mass induced by rainfall. A possible reason why two rock avalanches triggered in the Lushan earthquake travelled farther may be because of structural weakening of slope rock mass in the 2008 Wenchuan earthquake in the study area.

**5 Discussion**

**5.1 Prediction for travel distance of channelized rock avalanche**

The results of our analysis of the data set, indicates that the mobility (travel distance) of channelized rock avalanche is positively correlated with landslide volume and total relief but negatively correlated with channel angle. It is inferred that the movement of channelized rock avalanche was strictly constrained by the local geomorphology. As Figure 5 shows, the travel distance of channelized rock avalanche would rapidly increase with volume of rock avalanche enlarged. Such a high correlation between landslide volume and travel distance implies that the travel distance of channelized rock avalanche is dominated by the spreading of the slide mass (Davies, 1982; Staron,2009). The high

positive correlation between total relief and travel distance is for two reasons: the larger the total relief is, the more kinetic energy the slide mass could obtained and the further distance could it travel; another contribution is the geometrical similarity of hillslope geomorphology in the study area (Legros,

2002).

Regarding the medium negative correlation between travel distance and channel angle, it is inferred that when the slide mass rushed into the channel after the acceleration movement on the upper hillslope, it had relatively high velocity and extremely low frictional coefficient among the rock fragments, and the channel could not stop the rock avalanche until it lost fragment flow discharge. Hence, the travel distance of channelized rock avalanche would increase with the channel angle cut down given the same flow discharge (landslide volume), relative stable flow velocity, and similar flow capacity. However, it is still difficult to evaluate the flow capacity of the channels due to difficulty of quantifying its cross- section shape (width and depth of channels), resistance to the rock avalanche and even the shape changing induced by entrainment process of rock avalanche.

The residual analysis result demonstrates that the projection process in the early motion stage will significantly enlarge the travel distance of rock avalanches. The nature of this phenomenon is suggested to be involved with transformation of motion mode from sliding to flowing due to collision and fragmentation effects after the projection (Davies et al, 1999). Furthermore, the degree of fragmentation of failed mass should have remarkable influence on the travel distance of rock avalanche, and other factors changing the fragmentation degree should be further study, such as earthquake effect, geologic structure and rock type.

**5.2 Conceptual model for transportation of channelized rock avalanche**

The statistical results imply that the travel distance of channelized rock avalanche is highly correlated with landslide volume, total relief and channel angle. As the total relief and channel angle act as external factors for the motion of rock avalanche, it seems like it is in essence landslide volume that control the rock avalanche movement. Actually, a good fitting result between travel distance and landslide volume appears on our data set (Figure 4). So we propose a conceptual model for channelized rock avalanche transportation: An initial failed mass rushes into the channel with certain velocity after acceleration and fragmentation effects over the upper slope. Then the failed mass will "forget" the initial fall height and flow down in the channel like unsteady flow. The flow discharge (including

initial landslide volume and entrainment volume) and the flow capacity of the channel control the travel distance of channelized rock avalanche without considering the motion mechanism.

However, the flow capacity varies along the channel. Some local depression can store a mass of the moving rock debris, causing a lack of flow discharge for the downstream channel and a considerable decrease of travel distance. Taking Wenjia Gully rock avalanche for an example, almost a half of total volume of the landslide deposit on the beginning of the channel (red dash circle area in Figure 9), leading to that the distal deposition appeared in the channel instead of the valley. Thus assessing the flow capacity of the channel for rock avalanche motion will assist in future forecast of potential rock avalanche hazard in mountainous areas.

**6 Conclusion**

[revised manuscript text omitted]

Figure 9: Sketch map of flow capacity of channel affecting on the travel distance of Wenjia Gully channelized rock avalanche: (a) before the earthquake, (b) after the earthquake, (c) photo taken on deposition platform after the earthquake. The red arrow show the sliding direction of source mass. The red dotted line in figure.9(a) indicates the original depression on the travel path of the rock avalanche, in where debris deposition of about 30 million m3 was stored after the earthquake (shown in figure.9(b)), and more detailed information is shown in the figure.9(c).

[revised manuscript text omitted]

Note: The number in Italics indicates the two variables are not significantly correlated

**Table 3: The regression coefficients and results of significance tests of two multivariate regression models.**

| Equations | Coefficients* | Intercept | Coefficient of log(V) | Coefficient of log(H) | Coefficient of log(tanβ) | Coefficient of log(Hs) | Coefficient of log(tanβ) | Adjusted R² | F-stat | F0.05 |
|---|---|---|---|---|---|---|---|---|---|---|
| Best-fit regression equation | LCI | 0.175 | -0.013 | 0.521 | -0.548 | — | — |  |  |  |
|  | Mean | 0.420 | 0.079 | 0.718 | -0.365 | — | — | 0.933 | 173.5 | 2.883 |
|  | UCI | 0.665 | 0.171 | 0.914 | -0.182 | — | — |  |  |  |
| Alternative regression equation | LCI | 0.110 | 0.199 | — | -0.165 | -0.002 | -0.464 |  |  |  |
|  | Mean | 0.561 | 0.303 | — | 0.072 | 0.244 | -0.115 | 0.840 | 49.5 | 2.659 |
|  | UCI | 1.012 | 0.407 | — | 0.308 | 0.489 | 0.233 |  |  |  |

Note: "Coefficients" of each variable has three kinds: LCI is lower bound of the coefficients with 95% confidence; Mean is the mean value of the coefficients; UCI is upper bound of the coefficients with 95% confidence;

**Table 4: Background parameters and predicted values of 8 rock avalanches in the same area used for validation**

| Landslide name | Longitude | Latitude | Triggers* | V /10⁴m³ | B /° | α /° | Hs /m | H /m | L /m | L'(3)** /m | Error /% | L'(4)*** /m | Error /% |
|---|---|---|---|---|---|---|---|---|---|---|---|---|---|
| Pianqiaozi | 104.370 | 31.822 | WCEQ | 8.8 | 19 | 35 | 153 | 205 | 372 | 436 | 17.2 | 373 | 0.3 |
| Yangjiayan | 104.328 | 31.755 | WCEQ | 25.4 | 23 | 41 | 164 | 304 | 518 | 583 | 12.5 | 518 | 0.1 |
| Shanshulin | 103.508 | 31.181 | WCEQ | 27.9 | 25 | 34 | 340 | 433 | 715 | 731 | 2.3 | 660 | -7.6 |
| Fuyangou | 103.501 | 31.422 | WCEQ | 71.9 | 28 | 38 | 385 | 530 | 763 | 869 | 13.8 | 900 | 17.9 |

[Figure]

[Figure]

| Dayanbeng1 | 102.762 | 30.179 | LSEQ | 100 | 53 | 10 | 254 | 424 | 1267 | 1136 | -10.3 | 781 | -38.4 |
| Dayanbeng2 | 102.761 | 30.178 | LSEQ | 110 | 50 | 8 | 237 | 407 | 1372 | 1208 | -12.0 | 787 | -42.6 |
| Ermanshan | 102.739 | 29.322 | RF | 100 | 33 | 15 | 148 | 635 | 1370 | 1303 | -4.9 | 767 | -44.0 |
| Wulipo | 103.567 | 30.919 | RF | 150 | 30 | 10 | 135 | 377 | 1260 | 1078 | -14.4 | 833 | -33.9 |

Note: "Triggers" is the triggering condition of rock avalanches: "WCEQ" represents the 2008 Wenchuan Ms7.8 earthquake; "LSEQ" represents the 2013 Lushan Ms7.0 earthquake; "RF" represents the rock avalanche was induced by heavy rainfall.

$L'_{(3)}$, $L'_{(4)}$ indicates the predicted travel distance estimated by using equation (3) and (4) respectively.

---

## Author Comment (AC1) · 12 Mar 2017

Dear Prof. Mergili,

We acknowledge your time and helpful comments and advice very much, which are valuable for improving the quality of our manuscript. We have revised the manuscript carefully according to your and other reviewers' comments. The revised manuscript will be submitted after the reply to reviewers' comments stage. Your comments are reproduced below, followed by our responses and/or a summary of revisions to the manuscript in italic.

With my kindest regards, Sincerely, on behalf of my co-authors, Xuanmei Fan

[Figure]

Comments in general: Q1: Pages 3 and 4 are almost identical – I think that page 4 can just be deleted. R1: Thank you for point it out. Page 4 has been deleted.

Q2: A reference that could be interesting: Mergili, M., Krenn, J., Chu, H.-J. (2015) R2: We are sorry for missing this very interesting and relevant paper, which has been cited in the introduction section "Mergili et al. (2015) developed a multi-functional open source tool for backward- and forward-analyses of mass movement propagation".

Specific comments: We have done careful copy editing to revise grammar and style errors. Q3: Line 119: "topography" would be suitable rather than "geography" R3: We agree and it has been corrected to "Topography".

Q4: Line124: please explain what you mean with "slope transition angle" R4: We have explained it in the text. The slope transition angle refers to the angle between the failed upper slope and lower slope, which is the definition of Guo et al. (2014)

Q5: Line 130 what is the "angle of impact"? R5: We have rephrased the "angle of impact" to most commonly used term "angle of reach", which actually represents the relationship between the height of fall and maximum run-out distance, also called apparent coefficient of friction by Heim (1932).

Q6: Line 145 In many cases it is probably hard to clearly delineate the source area from the transition area – maybe you could shortly explain which strategy you applied to do so? R6: Thanks for your comments. For the channelized rock avalanches, their source area and transition area are somehow easy to be differentiated, as the source area are normally located at the top or upper part of slope, while the flow path (transition area) is partially or fully confined by channels. We added an explanation in the text: "The source area and the transition area of channelized rock avalanches in the study area were differentiated by their morphological characteristics observed first on remote sensing imagery and then checked in field."

Q7: Line 148–165: This part does NOT describe the data you use, but defines some

terms. It should be moved to the introduction. R7: Thank you for your comment, but we think this part fits better to the Data section, because it mainly defines the parameters in Table 1 that we used for building the regression models. Table 1 summarized the data from 38 channelized rock avalanches.

Q8: Line 159–160: Is L the Euclidean distance, or the distance along the flow path? R8: L is the Euclidean distance, which has been specified in the text. Q9: Line 176: You should give some examples or references demonstrating that the existing models did not produce a favourable prediction.

R9: We thank the reviewer for the nice comment. We will produce a new table to demonstrate this.

Q10: Line 182: You have to explain what "x" is in Eq. 1. R10: Thanks for pointing this out. We has specified that x (i= 1, 2, . . ., n) are the predictors ('independent variables', e.g. total relief, landslide volume etc.)

Q11: Line 238: Eq. 5 does not exist R11: This was a typo, which has been corrected.

Q12: Line 261: better use 103 or 106 instead of 104. R12: We have revised all the units to 103.

Q13: Line 296: What do you mean here with "projection"? R13: The projection process was a special type of failure mode of earthquake-triggered landslides that was first proposed by Huang et al. (2011). They defined that "ejection" (projection process here) as "Because of the landform enlargement effect of the earthquake wave, the slope close to earthquake fault or earthquake epicenter is pulled up from upper part or midupper part, and is thrown out, and forms projectile motion of the slope mass". Several features of the Wenchuan Earthquake had quite different characteristics from thoseproduced under general gravity force. Huang, R.Q., Xu, Q., Huo, J.J, 2011. Mechanism and Geo-mechanics Models of Landslides Triggered by 5.12 Wenchuan Earthquake. J.Mt.Sci 8:200-210

---

## Author Comment (AC2) · 17 Mar 2017

Dear Prof. Havenith,

We acknowledge your time and helpful comments, which are valuable for improving the quality of our manuscript. We have revised the manuscript carefully according to your and other reviewers' comments. The revised manuscript will be submitted after the reply to reviewers' comments stage. Your comments are reproduced below, followed by our responses and/or a summary of revisions to the manuscript.

With my kindest regards,

Sincerely, on behalf of my co-authors, Xuanmei Fan

Comments in general: Q1: Some essential aspects about the ratio of volume versus sliding surface are missing in the discussion and conclusions. You mainly considered the relatively classical parameters of volume (alone), slope angle, and total relief. The problem is treated as being almost 1D (linear along the slope) while channeling of rock avalanches is certainly also depending on the channel cross section and the presence of 'turns' along the channel. Those two aspects should be analysed as well. R1: We thank the reviewer for the insightful comments. We have focused on the maximum travel distance prediction of the rock avalanches in Wenchuan earthquake area. The effect of channel cross section and the channel direction turns on mass movement has been being always the challenge in landslide runout prediction, due to multi factors (e.g. channel geometry, material properties). We have address this issue in Section 5.2 to explain the channel geometry) in the revised version, please check the attached section 5.2.

Comments tied to sections: Q2: the specific conclusion of your work is missing. R2: Besides with the prediction models, we plan to draw the conclusion with the focus on influences of topography constrain, landslide types, material types (rock types), and triggers on the landslide mobility through the comparison with other datasets. These has been added in Section 5.2 of the revised version (see the attachment).

Detailed comments tied to lines: Q3: The grammar, terms and other similar details in lines 32, 62,139,156,195,297 R3: We have corrected the above lines.

Q4: Line 88-112; this page is the same as the previous one !!! To be deleted R4: We are sorry about this mistake. This page has been deleted.

Q5: Line 139: is that channel referring to the preexisting channel? R5: Yes. We have rephrased the 'channel' to 'pre-existing channel'.

Q6: Line 149: the concept of initiated slope (source area) R6: We have rephrased this as the source area means where the materials were initiated.

Q7: Line 155: repeat reference to fig. 3 where alpha and beta should be more high-lighted. R7: We have highlighted them.

Q8: Line 167: 'desirous' might not be used in this context. Maybe use 'most impor-tant prediction parameter'. R8: We have rephrased 'most desirous' to 'most important prediction parameter'.

Q9: Line 291: 'the travel distance of channelized rock avalanche would increase with the channel angle cut down given the same flow discharge' is not convincing ! The negative correlation with alpha is not logical and I think that it has an indirect effect ... meaning that some other parameter not studied here must explain this 'apparent' negative correlation. I think that the negative effect comes from the fact that the smaller alpha is compared to beta, the deeper is the sliding surface in the source area, - a very deep sliding surface ends up almost horizontal at the toe of the source area. This reduces alpha. R9: We agree with the reviewer and will add more explanation in the results part with taking more factors into consideration.

Q10: Line 293: the cross-section morphology was totally neglected in your analysis – why? Actually, the volume has this positive effect on travel distance as normally with larger volumes the volume-contact surface (reducing mobility due to friction) ratio increases. Additionally, curved cross-section profiles. up to a certain amount of curva-ture (typical for channels) reduce the total friction. For flat areas, the friction is highest as well as for very narrow channels with vertical walls. For medium curved channels the volume -contact surface ratio is lowest. R10: We agree with the reviewer. We have added some new figures to explain the correlation between volume and H/L ratio both from our dataset and world wide dataset. This will be discussed in detail in the revised version. We also would like to point out that the empirical-statistical method that we presented in this study only suits for the rapid assessment of potential runout of chan-nelized rock avalanches in the data-lack situation. The cross-section morphology could be obtained from DEM. However, in most case, DEM is not available. If it is available, numerical simulation using DEM could provide better results with consideration of the

detailed morphology than our method.

Comments on figures: Q11: Fig.3: (1) do not understand initiated slope. Maybe better: Failed upper part of the slope? (2) highlight better 'alpha' and 'beta' angles and refer to this figure when you use them in the equations. R11: We have revised Fig.3

Please also note the supplement to this comment:
http://www.nat-hazards-earth-syst-sci-discuss.net/nhess-2016-372/nhess-2016-372-AC2-supplement.pdf

**Supplement:**

**5.2 The mobility of channelized rock avalanche**

The mobility of landslides is influenced by varieties of factors, such as topography, landslide size, material type, landslide type water content and so on. The vital role of topography constrains on the landslide mobility can be referred from the high positive correlation of H/L with effective drop height, slope angle and channel angle (see Figure S2~S4). Besides, some micro topography like drop cliff and broad depression will influence the motion and deposition of rock avalanche remarkably. The rock avalanches corresponding with the four large bias scatter in Figure 8 are the Wenjia gully, Hongshi Gully, Niumian Gully and Donghekou rock avalanche, whose flow path has cliffs in the upper end of channels with notable drop height as 260 m, 150 m, 60 m and 160 m respectively referring to the field investigations. Moreover, fluidization characteristics such as super-elevation near curve transitions can be found in the channel section of these four rock avalanches. These findings manifest the steep micro-geotopography will enlarge the mobility of rock avalanches as this kind of topography will lead the slide mass to undergo the drop, collision, fragmentation effects in the early motion stage which will facilitate motion mode transformation from sliding to flowing. This transformation will enhance the motion mobility of rock avalanche to travel a much longer distance than predicted one. Attention also need be paid to the broad depression near the upper end of the channel. Taking Wenjia Gully rock avalanche for an example, almost a half of total volume of the landslide deposited on the beginning of the channel (red dash circle area in Figure 9), leading to that the travel distance lower than the expected one.

As for the effects of landslide types on the landslide mobility, we firstly compare our dataset with the dataset collected by Guo et al.(2014) in order to avoid the influences of triggers and topography. After the elimination of superposition parts between two datasets, 32 other landslides including debris avalanches, rock slides, soil slides in the same area are introduced. We plot the relationship between L with V and H respectively marking different types landslide (see figure S5 and S6). According to figure S5 and S6, rock avalanches show the strongest mobility while soil slides show the weakest one, and the mobility of rock slides is equable to the mobility of debris avalanches while the later one has large variation.

[Figure]

Figure S5 Relationship between the volume and travel distance of different-type landslides triggered by

Wenchuan earthquake (rock slides, debris avalanches and soil slides data are from Guo et al, 2014)

[Figure]

Figure S6 Relationship between total height and the travel distance of different-type landslides triggered

by Wenchuan earthquake (rock slides, debris avalanches and soil slides data are from Guo et al, 2014)

While compared with the worldwide datasets (see figure S7 and S8),

[Figure]

Figure S7 Relationship between the volume and H/L ratio of different-type landslides from the worldwide dataset

[Figure]

Figure S8 Relationship between the volume and H/L ratio of different-type landslides from the worldwide dataset

As for the effect of rock types on the landslide mobility, we use the combined dataset of landslides induced by Wenchuan earthquake. The rock type classification use the same standard used by Guo et al (refer to table 1 in Guo et al. 2014). The rock type is numbered by the sort from strongest to the weakest, namely R1 presents strongest rock type. According to figure S9 and S10, the mobility of landslide seems to increase with the increase of rock strength.

[Figure]

Figure S9 Relationship between the volume and travel distance of landslides with different rock types

[Figure]

Figure S10 Relationship between the total height and travel distance of landslides with different rock types

The common causes of landsides are earthquakes and rainfall. While the influences of triggers on landslide distribution is well studied, the effects of triggers on the landslide mobility is still a scientific gap. Zhang et al. (2013) indicated that rock avalanches triggered by earthquakes have slightly lower mobility than ones not triggered by earthquakes, and rock avalanches close to the seismic fault do not always have higher mobility even if a rock avalanche near the seismic fault is subjected to higher ground accelerations. Guo et al. (2014) also mentioned that the seismic acceleration plays less influence than rock type, sliding volume, slope transition angle and slope height on landslide travel distance. According to the table 4, two rainfall-induced rock avalanches show stronger mobility than earthquake-induced ones. The rock avalanches induced by rainfall express a stronger mobility than the earthquake-induced ones may due to lubrication effect of water. Hummocky surfaces are observed on the deposition of the Ermanshan rock avalanche. However, detailed study on the influence of triggers on the landslide mobility need further dataset.

---

## Author Comment (AC3) · 17 Mar 2017

Dear Prof. van Asch,

We acknowledge your time and helpful comments, which are valuable for improving the quality of our manuscript. We have revised the manuscript carefully according to your and other reviewers' comments. The revised manuscript will be submitted after the reply to reviewers' comments stage. Your comments are reproduced below, followed by our responses and/or a summary of revisions to the manuscript.

With my kindest regards, Sincerely, on behalf of my co-authors, Xuanmei Fan

Comments in general:

Q1: This is an interesting paper showing that with a limited amount of factors one is able to predict the travel distance of rock avalanches provided that they occur in the same area, are of the same type and have the same triggering conditions. This was already shown in this paper where the validation with landslides with other triggering conditions and lying in another area gave sometimes poor results. I am wondering why the authors did not mention in the introduction explicitly the use of the energy concept for runout modelling, which gives a simple transparent insight in the most important factors (relief and friction) influencing run-out distance. Interesting question arises also from the introduction about advantages and disadvantages of the use of deterministic physical models and statistical models.

R1: Thanks for your comments. We have added one chapter 4.1 to analyze the application of energy line model on the channelized landslides (see the revised text).

Comments tied to sections: Q2: In the introduction, the authors mention examples of important fast landslides but they must more precisely describe triggering condition and type.

R2: Thanks for your comment. We added a table summary of some commonly used empirical-statistical models for landslide motion prediction in Table S1 (please see the attachment 1).

Q3: I have great difficulty in presenting the total height (H) as an important factor for the run out distance since it is highly correlated with run-out distance (L) Therefore Equation 2 and 3 are really not useful predictive equations because you need the travel distance L which you have to predict? May be a trial an error procedure for L is a solution when using this equation? It would be nice to test this.

R3: We agree that travel distance (L) is highly correlated with total relief (H) partly due to the geomorphologic similarity in same area. Yet we suggest there are potential benefits of Eq(2) and Eq(3). As the Eq(2) is developed through a stepwise linear multivariate regression technique to get the best-fit multivariate regression model for travel
distance prediction, H is attracted to the equation when taking account of H, Hs, V, beta as input variables. On the physical base, total relief indicates the potential energy difference of the failure mass which control the motion of rock debris. Eq(2) can give a clue to the compare the influence of two important factors, volume and potential energy difference on the travel distance. From the point of practical use of Eq(2), the estimated results using Eq(2) can be a benchmark of the results obtained through other models especially while H can be estimated to close to elevation difference between source area and valley floor for cases with high possibility to reach the valley floor.

Q4: The authors solved the problem by making a correlation of Hs with H (Eq4) which is a practical solution but has of course no physical meaning and it has to be questioned whether it works in other areas. I want to see comments on this in the discussion paragraph.

R4: The positive correlation between Hs and L can be explained to be the increasing tendency of Hs with the volume.

Q5: The energy approach to model run-out (not used in this paper) shows that volume does not play a role. But in that case it is assumed that friction is not influenced by volume, which in practice seems to be the case due to all kinds of physical processes in the mass. Therefore in order to show this, I asked the authors to make also a correlation between H/L (mean friction during run-out) and volume.

R5: Thanks for your suggestion. We have added one chapter "4.1 Apparent coefficient of friction" to discuss the influence of several factors such volume, effective drop height, slope angle, channel angle on the H/L" (please check the attachment 2).

Q6: The effect of slope angle beta is a bit strange In Eq, 2 and 3 it is negative while in Eq 4 it is a positive factor. The authors should comment on this.

R6: Thanks for your comment. We suggest the positive maybe caused by the introduce of source area height and slope angle alfa to the regression model
Q7: The authors give sometimes unclear and peculiar explanations of their findings regarding the effect of volume on travel distance and the effect of total height and channel angle on run-out distance.

R7: We have clarified this in the revised version, which will be submitted before 20 April.

Q8: A lack of clarity for me sometimes occurred in the text where the authors give no definitions of some terms like flow capacity, projectile motion etc., (see my annotations and comments).

R8: Thanks for your comment. We explained these definitions in the relevant detailed comments tied to lines, see R32.

Detailed comments tied to lines:

Q9: The grammar, terms and other similar details in lines 40, 68, 76, 80, 85, 86, 138, 183, 184, 238, 241, 279, 317

R9: We have corrected the above lines.

Q10: Line 28: What is the role of water in these rock avalanches?

R10: The term "rock avalanche" has developed naturally in the literature, as a simplification of the complex "rock slide-debris avalanche", proposed by Varnes (1998). Hungr et al.,2001, suggested that the term "rock avalanche" be reserved for flow-like movements of fragmented rock resulting from major extremely rapid rock slides. This contrasts with the term "debris avalanche", which should be reserved for landslides originating in unconsolidated material. Therefore, the role of water on the motion of rock avalanches is omitted in this study.

Q11: Line 49: Make it more general: The statistical empirical models enveloped in one region cannot be applied in another region with different geomorphological and geological surroundings. And to be honest: the same holds nearly always for physical
models: due to the lag of parametric input data the parametric values have to be back calculated with passed events in a particular area and it has even to be seen whether these parametric values are valid for a next event in the same area

R11: We fully agree with the reviewer. We have added this in the discussion part of the revised version.

Q12: Three major sub-parallel faults were not marked in the map under this title.

R12: We are sorry about these mismatches of the sub-parallel faults. We have corrected the fault names in the text as "the Maoxian- Wenchuan fault, the Yingxiu-Beichuan fault and the Jiangyou-Guanxian fault".

Q13: Line 62: what are highly developed stream systems?

R13: We agree that the highly developed stream systems is not explicit. We have rephrased this sentence to 'With long-term endogenic and exogenic geological process, this region is characterized by high mountains and deep gorges and extreme rates of erosion'.

Q14: give an idea of the size of the fragments of these rock avalanche deposits.

R14: These rock avalanches deposits are mainly made up of debris with tens of centimeters as average particle size.

Q15: Rephrase the sentence 'When the source mass was detached from the slide bedrock, it may directly move into the channel down slope (see Figure 2b), or access the channel with enter it at some impact transition angle of movement direction (see Figure 2a)'.

R15: Thanks for this comment. We rephrase this paragraph to "The influence of the local geomorphology on the topography of the rock avalanche depositions can be recognized from remote-sensing images after the earthquake. The source area and the transition area of channelized rock avalanches in the study area were somehow easy

NHESSD
to be differentiated, as the source area are normally located at the top or upper part of slope, while the flow path (flow or transition area) is partially or fully confined by channels."

Q16: Line 88-112: Delete!! Repetition ! These section are a copy of what was printed above.

R16: We are sorry about this mistake. This page has been deleted.

Q17: Line 119: Vague! What means Geography in this case: 'Another well-known model is the statistical  $\alpha - \beta$  model in which the maximum runout distance is solely a function of geography'.

R17: We replaced "geography" with "topography".

Q18: In the first paragraph after chapter title 3.1 General consideration, namely line 116-127, indicate in the type of landslide which was investigated.

R18: Thanks for this comment. We added a table of published empirical relations related to landslide travel distance prediction, which summaries the keywords, model formula, type and trigger of landslide samples of different models (please check the attachment 1, Table S1).

Q19: Line 144-146: Altitude difference determines with mass the potential energy difference. The difference in potential energy is of course related to the travel distance but is not a deterministic factor. What surprises me is that in the fore going no attention was given explicitly to the energy method with the use of a friction lines to predict runout distances. That brings me to the question why the authors did not consider material type as a surrogate for friction.

R19: Thanks for your comment. We have added section 4.1 to analyze the apparent coefficient of friction of channelized rock avalanches and also done some comparison in the discussion, please check the attachment 2 for the new section 4.1.

NHESSD
Q20: Line 153-155: This is unclear: what is inclination of slope section and valley section. Why they are obtained. In the next sentence you talk about Slope angle (alpha) and Channel angle (beta) Is that the same as the inclinations mentioned in this highlighted sentence?

R20: Thanks for this comment. We aimed to explain the same thing with different expression. We agree that sentence is not clear and rephrase it to 'The average inclination of sectional slopes and channels are obtained respectively, while the gradient of valley section is neglected as it has very little variation.' The angles of slope and channel section are obtained and considered respectively to analyze the influences of different topographic conditions on the travel distance of rock avalanches.

Q21: Line 164: From a theoretical point of view the empirical link between area and volume is very tricky because rock strength of a failing block, and slope angle plays an important role in the depth of sliding and hence the volume. R21: We agree that the sliding depth and the volume are affected by the geological structure (like weak zone), topographic condition (like slope angle, location on the slope), groundwater level, ground motion intensity and so on. But there are several publication confirming that power-law equations indeed exists between the area and volume of landslides. Considering the difficulty of obtaining the volume of every rock avalanches due to the lack of pre-quake topographic data, we still regarded as a practical measure the relationship build with accordance to a popular volume-area relationship adapt by Guzzetti et al. (2009), Larsen et al. (2010) and calibrated with the field dataset in the Wenchuan earthquake area by Parker et al. (2011).

Larsen, I.J., Montgomery, D.R. and Korup, O., 2010. Landslide erosion controlled by hillslope material. Nature Geoscience, 3(4), pp.247-251.

Guzzetti, F., Ardizzone, F., Cardinali, M., Rossi, M. and Valigi, D., 2009. Landslide volumes and landslide mobilization rates in Umbria, central Italy. Earth and Planetary Science Letters, 279(3), pp.222-229.
Q22: Line 164-165: Unclear No idea what you mean: 'Volume of some rock avalanches with detailed field investigation are replaced by the data from published literature.'

R22: Thanks for the comment. We rephrased this sentence to 'The data of some rock avalanches with detailed field investigation are attained through the literature review.'

Q23: Line 173: But in that case the empirical-statistical methods may miss important factors when one does not knowing the physical processes of the mobility.

R23: We agree that some fundamental physical processes and principles should be considered during the empirical-statistical method construction. But as there are many unknowns related to the hypermobility of the rock avalanches, using empirical-statistical methods can considerably simplify the travel distance prediction. We rephrased this sentence to 'Empirical-statistical methods have long been used as tools to study the mobility of rock avalanche since they are easy to develop and apply, and not dependent on knowing the complex physical processes involved in the hypermobility of rock avalanches.'

Q24: Line 193: H is not independent of L.

R24: We agree with you about H is relevant with L and think it can be a basis of the regression model considering H as a variable at least from statistical view.

Q25: Line 198: The differences must have something to do with the difference in type of landslides. The here investigated landslides are all? rockslides triggered by the Earthquake fragmented into a rock avalanche

R25: We will find different datasets considering the landslide classification and then make more specific compare to analyze the influence of landslide types and topographic confinement on the motion ability of landslides. We have analyzed the influence of landslide types on the landslide mobility in the revised discussion part (see the attached new added section 5.2, attachment 3).

Q26: Line 209: It appears that in a basic energy approach for run out, volume is
canceled out and does not play a role if we assume that volume has no influence on the friction. But volume does have an influence on friction. Friction is lower at larger volumes which can be explained by all kind of physical processes. So it is nice to make a correlation between volume and H/L.

R26: We thank for your suggestion. We have made a new figure (Figure S7 in the attachment 3) to compare our dataset with the dataset of Legros et al. 2002. According this figure, the tendency that apparent friction angle (H/L) decreases with the increase of volume is still steady for channelized rock avalanches in our study. However, more scatters occur when the volume of channelized rock avalanches are less than approximately 4.0x106m3, which indicates topographic confinement may play a more important role than volume in determining the travel distance of landslides when the scale of landslide are relatively small.

Q27: Line 214 and 218: As the Eq.(2) and Eq.(3) show, if beta increases L decreases ?? R27: According to the results of best-fit regression and correlation analysis in Section 4.1, the L decreases with the increases of beta. Q28: Line 224: It looks to me also difficult to predict the area of rock mass which will fail?

R28: In our opinion, with adequate deformation premonition and detailed investigation, it is possible to reduce the uncertainty related to the scale estimation of potential slope failures.

Q29: Line 226: The correlation coefficient between H and alpha is not so high. Why do you want to introduce here alpha? In the foregoing you said that it is not a good correlator. Does it give a slighly better result with alpha in the equation?

R29: Yes it gives slightly better result.

Q30: Line 229: Compared with Eq 2 and 3 beta is now positive correlated with L in Eq 4??

R30: We will check this in the revised version.
Q31: Line 247: The Wenjia gully is of course a very complex one with among others a platform with a main deposition area half way.

R31: We agree with you. We want to take the Wenjia gully as an example to illustrate the influence of micro topography on the mobility of rock avalanches, especially of the broad depression at the upper end of channels.

Q32: Line 249: what is meaning of 'projectile motion' ?

R32: The projection process was a special type of failure mode of earthquake-triggered landslides that was first proposed by Huang et al. (2011). They defined that "ejection" (projection process here) as "Because of the landform enlargement effect of the earthquake wave, the slope close to earthquake fault or earthquake epicenter is pulled up from upper part or midupper part, and is thrown out, and forms projectile motion of the slope mass". Several features of the Wenchuan Earthquake had quite different characteristics from those produced under general gravity force. Projectile motion here mean the drop under the action of gravity only that the failure mass experienced when a steep and high slope is under the toe of source areas. The Donghekou landslide is a good example.

Huang, R.Q., Xu, Q., Huo, J.J, 2011. Mechanism and Geo-mechanics Models of Landslides Triggered by 5.12 Wenchuan Earthquake. J.Mt.Sci 8:200-210

Q33: Line 260: Are these validation landslides all rock slides transforming into debris avalanches?

R33: No, they are rock avalanches. Even though the last two avalanches were triggered by heavy rainfall, their motion did not have strong relations with water.

Q34: Line 265: With equation 4 we get large errors especially with the Lushan earthquake and when triggered by rain. Do we get an explanation?

R34: The significant underestimate of travel distance of rock avalanches triggered by the Lushan earthquake and heavy rainfall was supposed to be related to the decrease

**NHESSD**
of rock strength due to the Wenchaun earthquake.

Q35: Line 266: I find a 40 % error with Eq (4) a bit cumbersome. Maybe it have something to do with the trigger mechanism (rain) and another area (Lushan area more to the south).

R35: Yes, we have added some discussion on the effects of triggers on the landslide mobility. But in order to address this issue, further datasets are required.

Q36: Line 267: As for the best-fit regression model, But I am not so happy with the best fit regression model because it requires indirect knowledge of the predicted value (L) in order to obtain H.

R36: H could be considered as the vertical relief from the landslide source area to the nearest gully floor, which then could be obtained easily in the field or from the topography map.

Q37: Line 272: Was there an influence during the Wenchuan earthquake on rock weakening in the Lushan area?

R37: It is possible but we can not find enough evidence now. The influence of rock type (strength) is analyzed in the revised chapter 5.2.

Q38: Line 277: In Eq 2 and 3 beta is negatively correlated with L while in Eq 4 beta is positively correlated with L. I should expect that beta is always positively correlated with L.

R38: We will check this in the revised version.

Q39: Line 281-282: I do not see the logic of this. May be you can explain a bit more. It may have also something to do with a decrease in friction of larger volumes.

R39: 'Such a high correlation between landslide volume and travel distance implies that the travel distance of channelized rock avalanche is dominated by the spreading of the slide mass (Davies, 1982; Staron, 2009).'
Q40: Line 283-284: The kinetic energy varies along the track starting with zero to a maximum and ending with zero. The positive relation between H and L is determined by the friction line and the slope profile. The friction line start in the source area and crosses the slope in the lower part where the mass comes at rest A variation of slope profiles and a constant friction line will give a linear positive correlation between H and L The llinear correlation between H and L in Figure 3 shows that the friction is more or less constant around a mean value.

R40: We agree with the reviewer and will check this in the revised version.

Q41: Line 285: Unclear, need more explanation.

R41: We will check this in the revised version.

Q42: Line 287: the medium negative correlation between travel distance and channel angle was referred in chapter 4.1, but Eq 4 shows a positive correlation!

R42:The same as aforementioned comment. We will check this in the revised version.

Q43: Line 290: the sentence 'the channel could not stop the rock avalanche until it lost fragment flow discharge' is not clear.

R43: We have rewritten this discussion.

Q44: Line 291-292: If discharge and flow velocity are the same the crossectional flow area is the same. Width and depth of the crossection can change but what has that to do with a decreasing slope angle leading to a larger run-out distance? More explanation here. What do you mean by flow capacity?

R44: Thanks for your comment. We have rewritten this discussion.

Q45: Line 305: I am not so happy with the factor total relief because it is highly dependent on the run-out distance L. ' As the total relief and channel angle act as external factors for the motion of rock avalanche, it seems like it is in essence landslide volume that control the rock avalanche movement.'

**NHESSD**
R45: Thanks for your comment. We have rewritten this discussion.

Q46: Line 312: 'entrainment volume' is not considered in this paper Can be very important!

R46: Thanks for your comment. We have rewritten this discussion.

Q47: Line 312: you mean in our case Beta Because apart from volume and H nothing was considered in the equation 2 and 3 and in 4 Alpha and Hs. A bit confusing to introduce here the term flow capicity as a factor.

R47: Thanks for your comment. We have rewritten this discussion.

Q48: Line 318: 'leading to that the distal deposition appeared in the channel instead of the valley.'other (finer) half went into the valley.

R48: Thanks for your comment. We have rewritten this discussion. Comments on figures:

Q49: Fig 6, 7: the lateral axis titles are both log(L).

R49: Thanks for your comment. We have revised these two figures in the revised version.

Please also note the supplement to this comment:

http://www.nat-hazards-earth-syst-sci-discuss.net/nhess-2016-372/nhess-2016-372-AC3-supplement.pdf
needeelen paper

**Supplement:**

**Attachment 1**

**Table S1** Empirical-statistical models for landslide motion prediction

| General approach | Keywords to characterize the method | Triggering condition, Type | References |
|---|---|---|---|
| Travel angle | Volume $Log\ H/L=C_1 Log\ V+C_0$ | Rock fall/slide/avalanche debris flow/avalanche, earthflow | Scheidegger, 1973; Corominas, 1996 |
| | downslope angle $H/L=C_1\ tan\ S+C_0$ | soil slides, snow avalanches, nonseismic | *Hunter et al., 2003; Lied et al., 1980; McClung et al., 1987* |
| Total travel distance | Rock type, volume, slope transition angle $Log\ L=C_1 Rt+C_2 Log\ V+C_3\ sin\ S+C_0$ | rock/soil slide and rock/debris avalanche, earthquake | Guo et al., 2014 |
| | $Log\ L=C_1\ Log\ H+C_2\ Log\ tanS+C_0$ | Soil landslides, Artificial slopes | Finlay et al., 1999 |
| | $L=C_1 V^{C_2}$ | Debris slides, debris flowslides       rainfall | Jaiswal et al., 2011 |

Note: $C_0$, $C_1$, $C_2$, $C_3$ are the constants. L is the travel distance. H is the total height. V is the volume. S is the average slope angle. Rt is the rock type.

**Attachment 2**

**4.1 Apparent coefficient of friction**

Apparent coefficient of friction, also called the reach angle, is a well-known index to express the landslide mobility. It is the angle of the line connecting the crown of the landslide source area to the toe of the displaced mass. This angle is firstly conducted by Heim (1932) in the famous energy-line model as the average coefficient of friction of slide mass from initiation to rest. The apparent coefficient of friction is supposed to possess the ability of landslide mobility prediction because of its tendency to decrease with the increase of landslide volume illustrated by many researchers (Scheidegger, 1972; Corominas, 1996). Yet, the large scatter existed in these studies have impeded the application of apparent coefficient of friction. The ratio H/L is also queried by some researches to be physically meaningless (Legros, 2002)

In this study, the influence of landslide size, drop height, and terrain slope on the apparent coefficient of friction of the channelized rock avalanches are examined respectively (Figure S1 to S4). A negative correlation is between the Log10 volume and Log10 apparent coefficient of friction (See Figure S1) in accordance of historical studies (Scheidegger, 1972; Corominas, 1996). In order to consider the effect of potential energy on the H/L, the effective drop height defined as the total height minus the height of source area is used instead of the total height which excludes the superposition of source height and total height. That is especially useful for landslides with large-size initiation but limited travel distance. A significant positive correlation is observed between the H/L and effective drop height ignoring the four lower scatters in the Figure S2. From Figure S3 and S4, obvious positive correlation between the H/L and both the slope angle and channel angle can be determined. Although other robust evidences are missing, we suggest that the mobility (H/L) of channelized rock avalanches is controlled by local topography.

[Figure]

**Figure S1** Relationship between reach angle and volume

[Figure]

**Figure S2** Relationship between reach angle and effective drop height

[Figure]

**Figure S3** Relationship between reach angle and slope angle

[Figure]

**Figure S4** Relationship between reach angle and channel angle

**Attachment 3**

**5.2 The mobility of channelized rock avalanche**

The mobility of landslides is influenced by varieties of factors, such as topography, landslide size, material type, landslide type water content and so on. The vital role of topography constrains on the landslide mobility can be referred from the high positive correlation of H/L with effective drop height, slope angle and channel angle (see Figure S2~S4). Besides, some micro topography like drop cliff and broad depression will influence the motion and deposition of rock avalanche remarkably. The rock avalanches corresponding with the four large bias scatter in Figure 8 are the Wenjia gully, Hongshi Gully, Niumian Gully and Donghekou rock avalanche, whose flow path has cliffs in the upper end of channels with notable drop height as 260 m, 150 m, 60 m and 160 m respectively referring to the field investigations. Moreover, fluidization characteristics such as super-elevation near curve transitions can be found in the channel section of these four rock avalanches. These findings manifest the steep micro-geotopography will enlarge the mobility of rock avalanches as this kind of topography will lead the slide mass to undergo the drop,

collision, fragmentation effects in the early motion stage which will facilitate motion mode transformation from sliding to flowing. This transformation will enhance the motion mobility of rock avalanche to travel a much longer distance than predicted one. Attention also need be paid to the broad depression near the upper end of the channel. Taking Wenjia Gully rock avalanche for an example, almost a half of total volume of the landslide deposited on the beginning of the channel (red dash circle area in Figure 9), leading to that the travel distance lower than the expected one.

As for the effects of landslide types on the landslide mobility, we firstly compare our dataset with the dataset collected by Guo et al.(2014) in order to avoid the influences of triggers and topography. After the elimination of superposition parts between two datasets, 32 other landslides including debris avalanches, rock slides, soil slides in the same area are introduced. We plot the relationship between L with V and H respectively marking different types landslide (see figure S5 and S6). According to figure S5 and S6, rock avalanches show the strongest mobility while soil slides show the weakest one, and the mobility of rock slides is equable to the mobility of debris avalanches while the later one has large variation.

[Figure]

**Figure S5** Relationship between the volume and travel distance of different-type landslides triggered by Wenchuan earthquake (rock slides, debris avalanches and soil slides data are from Guo et al, 2014)

[Figure]

**Figure S6** Relationship between total height and the travel distance of different-type landslides triggered by Wenchuan earthquake (rock slides, debris avalanches and soil slides data are from Guo et al, 2014)

While compared with the worldwide datasets (see figure S7 and S8),

[Figure]

Figure S7 Relationship between the volume and H/L ratio of different-type landslides from the worldwide dataset

[Figure]

**Figure S8** Relationship between the volume and H/L ratio of different-type landslides from the worldwide dataset

As for the effect of rock types on the landslide mobility, we use the combined dataset of landslides induced by Wenchuan earthquake. The rock type classification use the same standard used by Guo et al (refer to table 1 in Guo et al. 2014). The rock type is numbered by the sort from strongest to the weakest, namely R1 presents strongest rock type. According to figure S9 and S10, the mobility of landslide seems to increase with the increase of rock strength.

[Figure]

**Figure S9** Relationship between the volume and travel distance of landslides with different rock types

[Figure]

**Figure S10** Relationship between the total height and travel distance of landslides with different rock types

The common causes of landsides are earthquakes and rainfall. While the influences of triggers on landslide distribution is well studied, the effects of triggers on the landslide mobility is still a scientific gap. Zhang et al. (2013) indicated that rock avalanches triggered by earthquakes have slightly lower mobility than ones not triggered by earthquakes, and rock avalanches close to the seismic fault do not always have higher mobility even if a rock avalanche near the seismic fault is subjected to higher ground accelerations. Guo et al. (2014) also mentioned that the seismic acceleration plays less influence than rock type, sliding volume, slope transition angle and slope height on landslide travel distance. According to the table 4, two rainfall-induced rock avalanches show stronger mobility than earthquake-induced ones. The rock avalanches induced by rainfall express a stronger mobility than the earthquake-induced ones may due to lubrication effect of water. Hummocky surfaces are observed on the deposition of the Ermanshan rock avalanche. However, detailed study on the influence of triggers on the landslide mobility need further dataset.

---

## Author Response (AR1)

**Cover letter to Editor:**

Dear Prof. Glade,

We acknowledge your time and the reviewers' helpful comments and advice very much, which are valuable for improving the quality of our manuscript. We reworked on the manuscript carefully by incorporating almost all the comments and suggestions that the reviewers suggested into the new version. **Two completely new sub-sections** have been added: subsection 4.1 "Reach angle of channelized rock avalanches" to illustrate the correlation between reach angle (H/L) and other parameters (i.e. volume, runout distance, the slope angle of the source area and the angle of the channel); and the subsection 5.2 "The mobility of channelized rock avalanches" to make a better and more interesting discussion. In subsection 5.2 we compared our data with the worldwide dataset and also the local dataset. In total, **5 new figures and 1 table** have been added. Therefore, to our best knowledge the manuscript has been largely improved. The reviewers' comments are reproduced below, followed by our responses and/or a summary of revisions to the manuscript in italic. A marked-up manuscript version with correction marked in red has also been attached at the end.

Sincerely, on behalf of my co-authors,
Xuanmei Fan

**Response to Reviewer Comments on Manuscript NHESS-2016-372:**

**Manuscript:** Empirical prediction for travel distance of channelized rock avalanches in the Wenchuan earthquake area

Weiwei Zhan, Xuanmei Fan, Runqiu Huang, Xiangjun Pei, Qiang Xu, Weile Li

**1. Response to Reviewer 1, Prof. Hans-Balder Havenith**

**Comments in general:**

**Q1:** Some essential aspects about the ratio of volume versus sliding surface are missing in the discussion and conclusions. You mainly considered the relatively classical parameters of volume (alone), slope angle, and total relief. The problem is treated as being almost 1D (linear along the slope) while channeling of rock avalanches is certainly also depending on the channel cross section and the presence of 'turns' along the channel. Those two aspects should be analysed as well.

*R1: We thank the reviewer for the insightful comments. The correlation between volume vs. H/L, slope angle of the source area (sliding surface) vs. H/L, and the channel angle along the flow path vs. H/L were presented in the new figures 4 and 5. We have focused on the maximum travel distance prediction of the rock avalanches in Wenchuan earthquake area. The effect of channel cross section and the channel direction turns on mass movement has been being always the challenge in landslide runout prediction, due to the factors (e.g. channel geometry, material properties). We have address this issue in Section 5.1 and the new Section 5.2 to explain the channel geometry) in the revised version (see lines 324-346).*

**Comments tied to sections:**

**Q2:** the specific conclusion of your work is missing.

*R2: Besides with the prediction models, we have added more discussions on the influences of topography constrain, landslide types, and triggers on the landslide mobility through the comparison with other datasets. These has been added in the newly added Section 5.2 of the revised version, see lines 314-361.*

**Detailed comments tied to lines:**

**Q3:** The grammar, terms and other similar details in lines 32, 62,139,156,195,297

*R3: We have corrected the above lines.*

**Q4:** Line 88-112; this page is the same as the previous one !!! To be deleted

*R4: We are sorry about this mistake. This page has been deleted.*

**Q5:** Line 139: is that channel referring to the preexisting channel?

*R5: Yes. We have rephrased the 'channel' to 'pre-existing channel'.*

**Q6:** Line 149: the concept of initiated slope (source area)

*R6: We have rephrased this as the source area means where the materials were initiated.*

**Q7:** Line 155: repeat reference to fig.3 where alpha and beta should be more highlighted.

*R7: We have highlighted them.*

**Q8:** Line 167: 'desirous' might not be used in this context. Maybe use 'most important prediction parameter'.

*R8: We have rephrased 'most desirous' to 'most important prediction parameter'.*

**Q9:** Line 291: 'the travel distance of channelized rock avalanche would increase with the channel angle cut down given the same flow discharge' is not convincing ! The negative correlation with alpha is not logical and I think that it has an indirect effect ... meaning that some other parameter not studied here must explain this 'apparent' negative correlation. I think that the negative effect comes from the fact that the smaller alpha is compared to beta, the deeper is the sliding surface in the source area, - a very deep sliding surface ends up almost horizontal at the toe of the source area. This reduces alpha.

*R9: We agree with the reviewer and has changed this part. See more discussions in section 5.1, lines 297-302, where we explained the reason of the negative correlation between travel distance and channel gradient. While in Equation 4 alpha has negative correlation with travel distance, but beta became positive. The reason for that might be the alpha somehow indirectly implies the depth of the source area and the corresponding volume.*

**Q10:** Line 293: the cross-section morphology was totally neglected in your analysis – why? Actually, the volume has this positive effect on travel distance as normally with larger volumes the volume-contact surface (reducing mobility due to friction) ratio increases. Additionally, curved cross-section profiles. up to a certain amount of curvature (typical for channels) reduce the total friction. For flat areas, the friction is highest as well as for very narrow channels with vertical walls. For medium curved channels the volume -contact surface ratio is lowest.

*R10: We agree with the reviewer. We have added some new figures to explain the correlation between volume and H/L ratio both from our dataset and world wide dataset*

*in the new subsection 5.2. We also would like to point out that the empirical-statistical method that we presented in this study only suits for the rapid assessment of potential runout of channelized rock avalanches in the data-lack situation. The cross-section morphology could be obtained from DEM. However, in most case, DEM is not available. If it is available, numerical simulation using DEM could provide better results with consideration of the detailed morphology than our method.*

**Comments on figures:**

**Q11:** Fig.3: (1) do not understand initiated slope. Maybe better: Failed upper part of the slope?

(2) highlight better 'alpha' and 'beta' angles and refer to this figure when you use them in the equations.

*R11: We have revised Fig.3*

**2. Response to Reviewer 2, Prof. Theo van Asch**

**Comments in general:**

**Q1:** This is an interesting paper showing that with a limited amount of factors one is able to predict the travel distance of rock avalanches provided that they occur in the same area, are of the same type and have the same triggering conditions. This was already shown in this paper where the validation with landslides with other triggering conditions and lying in another area gave sometimes poor results. I am wondering why the authors did not mention in the introduction explicitly the use of the energy concept for runout modelling, which gives a simple transparent insight in the most important factors (relief and friction) influencing run-out distance. Interesting question arises also from the introduction about advantages and disadvantages of the use of deterministic physical models and statistical models.

*R1: Thanks for your comments. We have added a new sub-section 4.1 to analyze the application of energy line model on the channelized landslides (see the revised text) and the new figures 4 and 5.*

**Comments tied to sections:**

**Q2:** In the introduction, the authors mention examples of important fast landslides but they must more precisely describe triggering condition and type.

*R2: Thanks for your comment. We added a new table to summarize some commonly used empirical-statistical models for landslide runout prediction in Table 1.*

**Q3:** lines 153-155: This is unclear: what is inclination of slope section and valley section. Why they are obtained. In the next sentence you talk about Slope angle (alpha) and Channel angle (beta) Is that the same as the inclinations mentioned in this highlighted sentence?

*R3: We have clarified this as "The average inclination of the source area and travel path are obtained respectively, while the gradient of valley floor (deposition area) is neglected as it has very little variation", which refer to "α" and "β" in the following sentence (see lines 136-138 in the revised version).*

**Q4:** lines 164-165: From a theoretical point of view the empirical link between area and volume is very tricky because rock strength of a failing block, and slope angle plays an important role in the depth of sliding and hence the volume.

*R4: It is tricky, but there are a lot of research making efforts on estimating volume using area, landslide type, rock type etc. as indicators to improve the statistic models. We agree that the sliding depth and the volume are affected by the geological structure (like weak zone), topographic condition (like slope angle, location on the slope), groundwater level, ground motion intensity and so on. But there are several publication confirming that power-law equations indeed exists between the area and volume of landslides. Considering the difficulty of obtaining the volume of every rock avalanches due to the lack of pre-quake topographic data, we still regarded as a practical measure the relationship build with accordance to a popular volume-area relationship adapt by Guzzetti et al. (2009), Larsen et al. (2010) and calibrated with the field dataset in the Wenchuan earthquake area by Parker et al. (2011) (see lines 141-142 in the revised version).*

*Larsen, I.J., Montgomery, D.R. and Korup, O., 2010. Landslide erosion controlled by hillslope material. Nature Geoscience, 3(4), pp.247-251.*

*Guzzetti, F., Ardizzone, F., Cardinali, M., Rossi, M. and Valigi, D., 2009. Landslide volumes and landslide mobilization rates in Umbria, central Italy. Earth and Planetary Science Letters, 279(3), pp.222-229.*

**Q5:** I have great difficulty in presenting the total height (H) as an important factor for the run out distance since it is highly correlated with run-out distance (L) Therefore Equation 2 and 3 are really not useful predictive equations because you need the travel distance L which you have to predict? May be a trial an error procedure for L is a solution when using this equation? It would be nice to test this.

*R5: We agree that travel distance (L) is highly correlated with total relief (H) partly due to the geomorphologic similarity in same area. Yet we suggest there are potential benefits of Eq(2) and Eq(3). As the Eq(2) is developed through a stepwise linear multivariate regression technique to get the best-fit multivariate regression model for travel distance prediction, H is attracted to the equation when taking account of H, Hs, V, beta as input variables. On the physical base, total relief indicates the potential energy difference of the failure mass which control the motion of rock debris. Eq(2) can give a clue to the compare the influence of two important factors, volume and potential energy difference on the travel distance. From the point of practical use of Eq(2), the estimated results using Eq(2) can be a benchmark of the results obtained through other models especially while **H can be estimated to close to elevation difference between source area and valley floor for cases with high possibility to reach the valley floor.***

**Q6:** The authors solved the problem by making a correlation of Hs with H (Eq4) which is a practical solution but has of course no physical meaning and it has to be questioned whether it works in other areas. I want to see comments on this in the discussion paragraph.

*R6: The positive correlation between Hs and L can be explained to be the increasing tendency of Hs with the volume.*

**Q7:** The energy approach to model run-out (not used in this paper) shows that volume does not play a role. But in that case it is assumed that friction is not influenced by volume, which in practice seems to be the case due to all kinds of physical processes in the mass. Therefore in order to show this, I asked the authors to make also a correlation between H/L (mean friction during run-out) and volume.

*R7: Thanks for your suggestion. We have added one chapter "4.1 Apparent coefficient of friction" to discuss the influence of several factors such volume, effective drop height, slope angle, channel angle on the H/L" (see lines 175-195).*

**Q8:** The effect of slope angle beta is a bit strange In Eq, 2 and 3 it is negative while in Eq 4 it is a positive factor. The authors should comment on this.

*R8: Thanks for your comment. We have discussed this in Section lines 297-302, as below*

*"The channel gradient is highly correlated with the H/L ratio as shown in Figure 5b, which actually represents the apparent friction coefficient along the flow path similar to the definition of angle of reach by Heim (1932). This is probably the reason of the negative correlation between travel distance and channel gradient, as the decrease of channel gradient means the decrease of static friction coefficient, and the increase of landslide volume and mobility (Figure 4a and Figure 12)."*

*This explains the negative influence of beta (the channel gradient) in Equation 2 and 3. While in Equation 4 beta becomes positive, it is probably due to the fact that alpha and beta together determine the reach angle (H/L). The positive maybe caused by the introduce of source area height and slope angle alpha to the regression model in Equation 4. Though this might be still not so convincing, this is what the data tell us. It is also possible some other site-specific factors played important role in controlling the landslide travel distance, but they could not be considered in the model, please see more discussions in the new section 5.2 (lines 314-361).*

**Q9:** The authors give sometimes unclear and peculiar explanations of their findings regarding the effect of volume on travel distance and the effect of total height and channel angle on run-out distance.

*R9: These have been clarified in Section 5.1*

**Q10:** A lack of clarity for me sometimes occurred in the text where the authors give no definitions of some terms like flow capacity, projectile motion etc., (see my annotations and comments).

*R10: Thanks for your comment. We explained these definitions in the relevant detailed comments tied to lines, see R32.*

**Detailed comments tied to lines:**

**Q11:** The grammar, terms and other similar details in lines 40, 68, 76, 80, 85, 86, 138, 183, 184, 238, 241, 279, 317

*R11: We have corrected the above lines.*

**Q12:** Line 28: What is the role of water in these rock avalanches?

*R12: The term "rock avalanche" has developed naturally in the literature, as a simplification of the complex "rock slide-debris avalanche", proposed by Varnes (1998). Hungr et al.,2001, suggested that the term "rock avalanche" be reserved for flow-like movements of fragmented rock resulting from major extremely rapid rock slides. This contrasts with the term "debris avalanche", which should be reserved for landslides originating in unconsolidated material. Therefore, the role of water on the motion of rock avalanches is omitted in this study.*

**Q13:** Line 49: Make it more general: The statistical empirical models enveloped in one region cannot be applied in another region with different geomorphological and geological surroundings. And to be honest: the same holds nearly always for physical models: due to the lag of parametric input data the parametric values have to be back calculated with passed events in a particular area and it has even to be seen whether these parametric values are valid for a next event in the same area

*R13: We fully agree with the reviewer. However, this is the problem of all these kinds of models, we could only wait to see whether it could really work for predicting the future events (even in other similar regions).*

**Q14:** Three major sub-parallel faults were not marked in the map under this title.

*R14: We are sorry about these mismatches of the sub-parallel faults. We have corrected the fault names in the text as "the Maoxian- Wenchuan fault, the Yingxiu-Beichuan fault and the Jiangyou-Guanxian fault".*

**Q15:** Line 62: what are highly developed stream systems?

*R15: We agree that the highly developed stream systems is not explicit. We have rephrased this sentence to 'With long-term endogenic and exogenic geological process, this region is characterized by high mountains and deep gorges with extreme rates of erosion'.*

**Q16:** Line 76: give an idea of the size of the fragments of these rock avalanche deposits.

*R16: These rock avalanches deposits are mainly made up of debris with tens of centimeters as average particle size. As we do not have exact grain size distribution data of all these rock avalanche deposition, we did not explain specific number here.*

**Q17:** Rephrase the sentence 'When the source mass was detached from the slide bedrock, it may directly move into the channel down slope (see Figure 2b), or access the channel with enter it at some impact transition angle of movement direction (see Figure 2a)'.

*R17: Thanks for this comment. We rephrase this paragraph to "The influence of the local geomorphology on the topography of the rock avalanche depositions can be recognized from remote-sensing images after the earthquake. The source area and the transition area of channelized rock avalanches in the study area were somehow easy to be differentiated, as the source area are normally located at the top or upper part of slope, while the flow path (flow or transition area) is partially or fully confined by channels." (in lines 86-90)*

**Q18:** Line 88-112: Delete!! Repetition ! These section are a copy of what was printed above.

*R18: We are sorry about this mistake. This page has been deleted.*

**Q19:** Line 119: Vague! What means Geography in this case: 'Another well-known model is the statistical α–β model in which the maximum runout distance is solely a function of geography'.

*R19: We replaced "geography" with "topography".*

**Q20:** In the first paragraph after chapter title 3.1 General consideration, namely line 116-127, indicate in the type of landslide which was investigated.

*R20: Thanks for this comment. We added a table of published empirical relations related to landslide travel distance prediction, which summaries the keywords, model formula, type and trigger of landslide samples of different models (Table 1 in the revised version).*

**Q21:** Line 144-146: Altitude difference determines with mass the potential energy difference. The difference in potential energy is of course related to the travel distance but is not a deterministic factor. What surprises me is that in the fore going no attention was given explicitly to the energy method with the use of a friction lines to predict run-out distances. That brings me to the question why the authors did not consider material type as a surrogate for friction.

*R21: Thanks for your comment. We have added section 4.1 (lines 177-196) to analyze the apparent coefficient of friction of channelized rock avalanches and also done some comparison in the discussion part.*

**Q22:** Line 164-165: Unclear No idea what you mean: 'Volume of some rock avalanches with detailed field investigation are replaced by the data from published literature.'

*R22: Thanks for the comment. We rephrased this sentence to "For some rock avalanches with field measured volume available, we use field measurement data rather than the estimated volume by area" in lines 147-148.*

**Q23:** Line 173: But in that case the empirical-statistical methods may miss important factors when one does not knowing the physical processes of the mobility.

*R23: We agree that some fundamental physical processes and principles should be considered during the empirical-statistical method construction. But as there are many unknowns related to the hypermobility of the rock avalanches, using empirical-statistical methods can considerably simplify the travel distance prediction. We rephrased this sentence to 'Empirical-statistical methods have long been used as tools to study the mobility of rock avalanche since they are easy to develop and apply, and not dependent on knowing the complex physical processes involved in the hypermobility of rock avalanches.'*

**Q24:** Line 193: H is not independent of L.

*R24: We agree with you about H is relevant with L and think it can be a basis of the regression model considering H as a variable at least from statistical view.*

**Q25:** Line 198: The differences must have something to do with the difference in type of landslides. The here investigated landslides are all? rockslides triggered by the Earthquake fragmented into a rock avalanche

*R25: We have found different datasets considering the landslide classification and then make more specific compare to analyze the influence of landslide types and topographic confinement on the motion ability of landslides. We have analyzed the influence of landslide types on the landslide mobility in the discussion part section 5.2 in lines 324-374.*

**Q26:** Line 209: It appears that in a basic energy approach for run out, volume is canceled out and does not play a role if we assume that volume has no influence on the friction. But volume does have an influence on friction. Friction is lower at larger volumes which can be explained by all kind of physical processes. So it is nice to make a correlation between volume and H/L.

*R26: We thank for your suggestion. We have made a new figure (Figure 13 in the revised version) to compare our dataset with the dataset of Legros et al. (2002). According this figure, the tendency that apparent friction angle (H/L) decreases with the increase of volume is still steady for channelized rock avalanches in our study. However, more scatters occur when the volume of channelized rock avalanches are less than approximately $4.0x10^6m^3$, which indicates topographic confinement may play a more important role than volume in determining the travel distance of landslides when the scale of landslide are relatively small.*

**Q27:** Line 214 and 218: As the Eq.(2) and Eq.(3) show, if beta increases L decreases ??

*R27: Please see the answer to Q8 in R8.*

**Q28:** Line 224: It looks to me also difficult to predict the area of rock mass which will fail?

*R28: In our opinion, with adequate deformation premonition and detailed investigation, it is possible to reduce the uncertainty related to the scale estimation of potential slope failures.*

**Q29:** Line 226: The correlation coefficient between H and alpha is not so high. Why do you want to introduce here alpha? In the foregoing you said that it is not a good correlator. Does it give a slighly better result with alpha in the equation?

*R29: Yes it gives slightly better result.*

**Q30:** Line 229: Compared with Eq 2 and 3 beta is now positive correlated with L in Eq 4??

**R30:** *Please see the answer to Q8 in R8.*

**Q31:** Line 247: The Wenjia gully is of course a very complex one with among others a platform with a main deposition area half way.

**R31***: We agree with you. We want to take the Wenjia gully as an example to illustrate the influence of micro topography on the mobility of rock avalanches, especially of the broad depression at the upper end of channels.*

**Q32:** Line 249: what is meaning of 'projectile motion' ?

**R32***: The projection process was a special type of failure mode of earthquake-triggered landslides that was first proposed by Huang et al. (2011).* The projection phenomenon was observed in the Wenchuan earthquake region by Huang et al. (2011), defined us the thrown out or projectile motion of slope material due to site amplification effect of seismic wave causing the PGA large than 1 g (lines 314-316). Several features of the Wenchuan Earthquake had quite different characteristics from those produced under general gravity force. The Donghekou landslide is a good example.

*Huang, R.Q., Xu, Q., Huo, J.J, 2011. Mechanism and Geo-mechanics Models of Landslides Triggered by 5.12 Wenchuan Earthquake. J.Mt.Sci 8:200-210*

**Q33:** Line 260: Are these validation landslides all rock slides transforming into debris avalanches?

**R33***: No, they are rock avalanches. Even though the last two avalanches were triggered by heavy rainfall, their motion did not have strong relations with water.*

**Q34:** Line 265: With equation 4 we get large errors especially with the Lushan earthquake and when triggered by rain. Do we get an explanation?

**R34***: The significant underestimate of travel distance of rock avalanches triggered by the Lushan earthquake and heavy rainfall was supposed to be related to the decrease of rock strength due to the Wenchaun earthquake.*

**Q35:** Line 266: I find a 40 % error with Eq (4) a bit cumbersome. Maybe it have something to do with the trigger mechanism (rain) and another area (Lushan area more to the south).

**R35:** *Yes, we have added some discussion on the effects of triggers on the landslide mobility. But in order to address this issue, further datasets are required, see lines 283-288.*

**Q36:** Line 267: As for the best-fit regression model, But I am not so happy with the best fit regression model because it requires indirect knowledge of the predicted value (L) in order to obtain H.

**R36:** *H could be considered as the vertical relief from the landslide source area to the nearest gully floor, which then could be obtained easily in the field or from the topography map. Therefore, the results calculated through Eq.2 are possible used as a preliminary estimation of the rock avalanche travel distance.*

**Q37:** Line 272: Was there an influence during the Wenchuan earthquake on rock weakening in the Lushan area?

**R37:** *It is possible but we could not find enough evidence now.*

**Q38:** Line 277: In Eq 2 and 3 beta is negatively correlated with L while in Eq 4 beta is positively correlated with L. I should expect that beta is always positively correlated with L.

**R38:** *Please see the answer to Q8 in R8.*

**Q39:** Line 281-282: I do not see the logic of this. May be you can explain a bit more. It may have also something to do with a decrease in friction of larger volumes.

**R39:** *'Such a high correlation between landslide volume and travel distance implies that the travel distance of channelized rock avalanche is dominated by the spreading of the slide mass (Davies, 1982; Staron,2009).'*

**Q40:** Line 283-284: The kinetic energy varies along the track starting with zero to a maximum and ending with zero. The positive relation between H and L is determined by the friction line and the slope profile. The friction line start in the source area and crosses the slope in the lower part where the mass comes at rest A variation of slope profiles and a constant friction line will give a linear positive correlation between H and L The llinear correlation between H and L in Figure 3 shows that the friction is more or less constant around a mean value.

**R40:** *We agree with the reviewer, see more discussions in sub-section 5.2 an 4.1.*

**Q41:** Line 285: Unclear, need more explanation.

*R41: We have clarified this in the revised version.*

**Q42:** Line 287: the medium negative correlation between travel distance and channel angle was referred in chapter 4.1, but Eq 4 shows a positive correlation!

*R42: Please see the answer to Q8 in R8.*

**Q43:** Line 290: the sentence 'the channel could not stop the rock avalanche until it lost fragment flow discharge' is not clear.

*R43: We have deleted that part, please see the new section 5.2 (line 314-361)*

**Q44:** Line 291-292: If discharge and flow velocity are the same the crossectional flow area is the same . Width and depth of the crossection can change but what has that to do with a decreasing slope angle leading to a larger run-out distance? More explanation here. What do you mean by flow capacity?

*R44: We have deleted that part, please see the new section 5.2 (line 314-361)*

**Q45:** Line 305: I am not so happy with the factor total relief because it is highly dependent on the run-out distance L. ' As the total relief and channel angle act as external factors for the motion of rock avalanche, it seems like it is in essence landslide volume that control the rock avalanche movement.'

*R45: please see R36, which is the same question.*

**Q46:** Line 312: 'entrainment volume' is not considered in this paper Can be very important!

*R46: We agree, however without detailed pre- and post-event DEMs, it is not possible to quantify the entrainment volume.*

**Q47:** Line 312: you mean in our case Beta Because apart from volume and H nothing was considered in the equation 2 and 3 and in 4 Alpha and Hs. A bit confusing to introduce here the term flow capacity as a factor.

*R47: Please see the answer to Q8 in R8.*

**Comments on figures:**

**Q48:** Fig 6, 7: the lateral axis titles are both log(L).

*R48: Thanks for your comment. We have revised these two figures in the revised version.*

**3. Response to Reviewer 3, Prof. Martin Mergili**

**Comments in general:**

**Q1:** Pages 3 and 4 are almost identical – I think that page 4 can just be deleted.

*R1: Thank you for point it out. Page 4 has been deleted.*

**Q2:** A reference that could be interesting: Mergili, M., Krenn, J., Chu, H.-J. (2015)

*R2: We are sorry for missing this very interesting and relevant paper, which has been cited in the introduction section "Mergili et al. (2015) developed a multi-functional open source tool r.randomwalk, for conceptual modelling of the propagation of mass movements, which combined the empirical model with the numerical model." (lines 44-46)*

**Specific comments:**

*We have done careful copy editing to revise grammar and style errors.*

**Q3:** Line 119: "topography" would be suitable rather than "geography"

*R3: We agree and it has been corrected to "Topography".*

**Q4:** Line124: please explain what you mean with "slope transition angle"

*R4: We have explained it in the text. The slope transition angle refers to the angle between the failed upper slope and lower slope, which is the definition of Guo et al. (2014), see line 103 in the revised version.*

**Q5:** Line 130 what is the "angle of impact"?

*R5: We have rephrased the "angle of impact" to most commonly used term "angle of reach" in line 111, which actually represents the relationship between the height of fall*

*and maximum run-out distance, also called apparent coefficient of friction by Heim (1932).*

**Q6:** Line 145 In many cases it is probably hard to clearly delineate the source area from the transition area – maybe you could shortly explain which strategy you applied to do so?

*R6: Thanks for your comments. For the channelized rock avalanches, their source area and transition area are somehow easy to be differentiated, as the source area are normally located at the top or upper part of slope, while the flow path (transition area) is partially or fully confined by channels. We added an explanation in the text: "The source area and the transition area of channelized rock avalanches in the study area were somehow easy to be differentiated, as the source area are normally located at the top or upper part of slope, while the flow path (flow or transition area) is partially or fully confined by channels" (in lines 87-90).*

**Q7:** Line 148–165: This part does NOT describe the data you use, but defines some terms. It should be moved to the introduction.

*R7: Thank you for your comment, but we think this part fits better to the Data section, because it mainly defines the parameters in Table 1 that we used for building the regression models. Table 1 summarized the data from 38 channelized rock avalanches.*

**Q8:** Line 159–160: Is L the Euclidean distance, or the distance along the flow path?

*R8: L is the Euclidean distance, which has been specified in the text (line 142).*

**Q9:** Line 176: You should give some examples or references demonstrating that the existing models did not produce a favourable prediction.

*R9: We thank the reviewer for the nice comment. We have added Fig.13 to show the difference of our dataset with others' models. In addition, a new table (Table 1) has been added to summarize the existing models in the literature.*

**Q10:** Line 182: You have to explain what "x" is in Eq. 1.

**R10:** *Thanks for pointing this out. We has specified that x (i= 1, 2, …, n) are the predictors ('independent variables', e.g. total relief, landslide volume etc, see line 170-171.*

**Q11:** Line 238: Eq. 5 does not exist

**R11:** *This was a typo, which has been corrected.*

**Q12:** Line 261: better use $10^3$ or $10^6$ instead of $10^4$.

**R12:** *We have revised all the units to $10^3$.*

**Q13:** Line 296: What do you mean here with "projection"?

**R13:** *The projection process was a special type of failure mode of earthquake-triggered landslides that was first proposed by Huang et al. (2011). 
[revised manuscript text omitted]

Note: The number in Italics indicates the two variables are not significantly correlated

**Table 4** The regression coefficients and results of significance tests of two multivariate regression models

| Equations | Coefficients* | Intercept | Coefficient of log(V) | Coefficient of log(H) | Coefficient of log(tanβ) | Coefficient of log(Hs) | Coefficient of log(tanβ) | Adjusted R² | F-stat | F0.05 |
|---|---|---|---|---|---|---|---|---|---|---|
| Best-fit regression equation | LCI | 0.175 | -0.013 | 0.521 | -0.548 | — | — |  |  |  |
|  | Mean | 0.420 | 0.079 | 0.718 | -0.365 | — | — | 0.933 | 173.5 | 2.883 |
|  | UCI | 0.665 | 0.171 | 0.914 | -0.182 | — | — |  |  |  |
| Alternative regression equation | LCI | 0.110 | 0.199 | — | -0.165 | -0.002 | -0.464 |  |  |  |
|  | Mean | 0.561 | 0.303 | — | 0.072 | 0.244 | -0.115 | 0.840 | 49.5 | 2.659 |
|  | UCI | 1.012 | 0.407 | — | 0.308 | 0.489 | 0.233 |  |  |  |

Note: "Coefficients" of each variable has three kinds: LCI is lower bound of the coefficients with 95% confidence; Mean is the mean value of the coefficients; UCI is upper bound of the coefficients with 95% confidence;

**Table 5** Background parameters and predicted values of 8 rock avalanches in the same area used for validation

| Landslide name | Longitude | Latitude | Triggers* | V /10⁴m³ | α /° | B /° | Hs /m | H /m | L /m | L'(3)** /m | Error / % | L'(4)*** /m | Error / % |
|---|---|---|---|---|---|---|---|---|---|---|---|---|---|
| Pianqiaozi | 104.370 | 31.822 | WCEQ | 8.8 | 35 | 19 | 153 | 205 | 372 | 436 | 17.2 | 373 | 0.3 |
| Yangjiayan | 104.328 | 31.755 | WCEQ | 25.4 | 41 | 23 | 164 | 304 | 518 | 583 | 12.5 | 518 | 0.1 |
| Shanshulin | 103.508 | 31.181 | WCEQ | 27.9 | 34 | 25 | 340 | 433 | 715 | 731 | 2.3 | 660 | -7.6 |
| Fuyangou | 103.501 | 31.422 | WCEQ | 71.9 | 38 | 28 | 385 | 530 | 763 | 869 | 13.8 | 900 | 17.9 |

| | | | | | | | | | | | | | |
|---|---|---|---|---|---|---|---|---|---|---|---|---|---|
| Dayanbeng1 | 102.762 | 30.179 | LSEQ | 100 | 53 | 10 | 254 | 424 | 1267 | 1136 | -10.3 | 781 | -38.4 |
| Dayanbeng2 | 102.761 | 30.178 | LSEQ | 110 | 50 | 8 | 237 | 407 | 1372 | 1208 | -12.0 | 787 | -42.6 |
| Ermanshan | 102.739 | 29.322 | RF | 100 | 33 | 15 | 148 | 635 | 1370 | 1303 | -4.9 | 767 | -44.0 |
| Wulipo | 103.567 | 30.919 | RF | 150 | 30 | 10 | 135 | 377 | 1260 | 1078 | -14.4 | 833 | -33.9 |

Note: "Triggers" is the triggering condition of rock avalanches: "WCEQ" represents the 2008 Wenchuan $M_s$ 8.0 earthquake; "LSEQ" represents the 2013 Lushan $M_s$ 7.0 earthquake; "RF" represents the rock avalanche was induced by heavy rainfall. $L'_{(3)}$, $L'_{(4)}$ indicates the predicted travel distance estimated by using equation (3) and (4) respectively.

[Figure]

**Figure 1.** Distribution map of large rock avalanches triggered by the Wenchuan earthquake

[Figure]

**Figure 2.** Remote-sensing images of two channelized rock avalanches triggered by the Wenchuan earthquake. a is Changtan rock avalanche (No.21 in table 2); b is Laoyingyan rock avalanche, which is river-blocked

[Figure]

**Figure 3.** Sketch map of a channelized rock avalanche defining geometric parameters. The red-dashed ellipse indicates the topographic transition dividing the initiated slope, channel and valley floor. The red arrow represents sliding direction of source mass

[Figure]

**Figure 4.** (a) Relationship between reach angle (H/L) and volume (V); and (b) Relationship between H/L and effective drop height of channelized rock avalanches (H-Hs).

[Figure]

**Figure 5.** (a) Relationship between reach angle (H/L) and slope angle (tan α); and (b) Relationship between H/L and the channel gradient (tan β) of the rock avalanches

[Figure]

**Figure 6.** Relationship between horizontal travel distance and volume of channelized rock avalanches

[Figure]

**Figure 7.** Relationship between horizontal travel distance and total relief of channelized rock avalanches

[Figure]

**Figure 8**. Residual plots for the two multivariate regression models: Figure 9a is for equation (2); Figure 9b is for equation (4).

[Figure]

**Figure 9.** The comparison between observed and predicted travel distance for the two multivariate regression models

[Figure]

**Figure 10.** Sketch map of flow capacity of channel affecting on the travel distance of the Wenjia Gully channelized rock avalanche: (a) before the earthquake, (b) after the earthquake, (c) photo taken on deposition platform after the earthquake. The red arrow shows the sliding direction of source mass. The red dotted line in (a) indicates the original depression on the travel path of the rock avalanche, in where debris deposition of about 30 million m3 was stored after the earthquake (shown in (b)), and more detailed information is shown in (c)

[Figure]

**Figure 11.** Relationship between the volume and travel distance (a), as well as relationship between the total height and travel distance (b) of different-type landslides triggered by Wenchuan earthquake (rock slides, debris avalanches and soil slides data are from Guo et al, 2014).

[Figure]

**Figure 12**. Relationship between the volume and H/L ratio of different-type landslides from the worldwide dataset (Corominas, 1996)

[Figure]

**Figure 13.** Relationship between the volume and H/L ratio of different-type landslides from the worldwide dataset (Legros, 2002)